# Effects of presenilin-1 familial Alzheimer's disease mutations on γ-secretase activation for cleavage of amyloid precursor protein

Hung N. Do [1,3], Sujan Devkota[2,3], Apurba Bhattarai[1], Michael S. Wolfe [2✉] & Yinglong Miao [1✉]

Presenilin-1 (PS1) is the catalytic subunit of γ-secretase which cleaves within the transmembrane domain of over 150 peptide substrates. Dominant missense mutations in PS1 cause early-onset familial Alzheimer's disease (FAD); however, the exact pathogenic mechanism remains unknown. Here we combined Gaussian accelerated molecular dynamics (GaMD) simulations and biochemical experiments to determine the effects of six representative PS1 FAD mutations (P117L, I143T, L166P, G384A, L435F, and L286V) on the enzyme-substrate interactions between γ-secretase and amyloid precursor protein (APP). Biochemical experiments showed that all six PS1 FAD mutations rendered γ-secretase less active for the endoproteolytic (ε) cleavage of APP. Distinct low-energy conformational states were identified from the free energy profiles of wildtype and PS1 FAD-mutant γ-secretase. The P117L and L286V FAD mutants could still sample the "Active" state for substrate cleavage, but with noticeably reduced conformational space compared with the wildtype. The other mutants hardly visited the "Active" state. The PS1 FAD mutants were found to reduce γ-secretase proteolytic activity by hindering APP residue L49 from proper orientation in the active site and/or disrupting the distance between the catalytic aspartates. Therefore, our findings provide mechanistic insights into how PS1 FAD mutations affect structural dynamics and enzyme-substrate interactions of γ-secretase and APP.

[1] Center for Computational Biology and Department of Molecular Biosciences, University of Kansas, Lawrence, KS 66047, USA. [2] Department of Medicinal Chemistry, School of Pharmacy, University of Kansas, Lawrence, KS 66047, USA. [3] These authors contributed equally: Hung N. Do, Sujan Devkota. ✉email: mswolfe@ku.edu; miao@ku.edu

γ-Secretase is an intramembrane aspartyl protease complex composed of four components: Nicastrin (NCT), Aph-1, Pen-2, and Presenilin-1 (PS1)[1,2]. PS1 is the catalytic component of γ-secretase, "the proteasome of the membrane"[3] which carries out intramembrane proteolysis of more than 150 peptide substrates[4], including amyloid precursor protein (APP), via two conserved aspartates, D257 and D385[5,6]. Dominant missense mutations in PS1 can cause early-onset familial Alzheimer's disease (FAD), a deadly chronic neurodegenerative disorder[7]. Although disease-causing PS1 mutations were first identified over 25 years ago, exact pathogenic mechanisms of FAD mutations remain unclear.

Two primary hypotheses have been proposed to explain the pathogenesis of FAD mutations. The loss-of-function hypothesis contends that PS1 FAD mutations reduce proteolytic activity of γ-secretase, which would impair cell signaling pathways by interfering with normal physiological functions of cleavage products, thereby leading to memory impairment and neurodegeneration[8–10]. In contrast, the gain-of-function hypothesis states that most FAD mutations increase the production of longer, more aggregation-prone Aβ peptides, resulting in toxic oligomers that trigger Alzheimer's disease (AD)[10–12]. However, these apparently opposing hypotheses can be reconciled by our experimental findings showing that PS1 FAD-mutant γ-secretase complexes are dramatically deficient in tricarboxypeptidase trimming of Aβ49 and Aβ48 initially produced through endoproteolytic (ε) cleavage[13,14]. Reduced trimming was also recently seen with 14 different FAD mutations in APP[15]. These reduced trimmings results in increased ratios of 42-residue Aβ (Aβ42)—the primary component of AD cerebral plaques—to Aβ40[14,16] as well as increased proportions of longer intermediates Aβ45–Aβ49[13,14]. Recently, Sun et al. analyzed 138 pathogenic mutations in the PS1 of γ-secretase on the in vitro production of Aβ42 and Aβ40 peptides[17]. They found that ~90% of the mutations reduced the production of Aβ42 and Aβ40, and ~10% of these mutations decreased the Aβ42/Aβ40 ratio[17]. Moreover, Trambauer et al. studied seven Aβ43-producing PS1 FAD mutants, including M292D, L166P, V261F, Y256S, R278I, G382A, and L435F, and found that Aβ43 was produced in very high levels when the PS1 function was severely impaired[18]. Furthermore, alteration of enzyme-C99-substrate interactions were observed in all these mutants, regardless of their effects[18].

Molecular dynamics (MD) is a powerful computational technique for simulating biomolecular dynamics at an atomistic level[19]. Kong et al.[20] performed the first atomistic simulation of isolated PS1 unit in 2015 and found that transmembrane domains (TM) 2, 6, and 9 were highly mobile[21,22]. In addition, only inactive distances between catalytic aspartates were sampled in the study because of the electrostatic repulsion caused by the negative charges of the two aspartates forming the active site[20,21]. The coarse-grained simulations of PS1 as part of the γ-secretase complex illustrated that PS1 was much more likely to be activated when either of the catalytic aspartates was protonated[23]. This finding was in good agreement with the proposed mechanism of aspartic proteases, which requires one of the catalytic aspartates to act as an acid[24]. Hitzenberger and Zacharias observed that the active state of PS1 remained stable even in the absence of a substrate as the direct hydrogen bond between protonated D257, D385, and a water bridge was sufficient to stabilize the active form[21,25]. Furthermore, the transition towards the active state of PS1 was found to involve TM1, TM6, TM7, TM8, and TM9[21,25]. In one recent study, conventional MD (cMD) has been applied to simulate the PS1 FAD mutations of E280A, G384A, A434C, and L435F, and APP FAD mutation of V717I. The simulations suggested that FAD

mutations destabilize the enzyme-substrate complexes[10]. However, both catalytic aspartates were deprotonated in the system setups, likely resulting in repulsion between the negative charges. The enzyme thus could not become active for substrate proteolysis during the simulations. In another study, free energy simulations have been carried out to examine the effects of selected PS1 FAD mutations, including L250S, S390I, L392V, L435S, P436S, and I439V[26]. Although different free energy profiles were revealed for the FAD mutants compared with the wildtype, these simulations were carried out in the absence of the substrate and the effects of FAD mutations on enzyme-substrate interactions could not be explored.

Gaussian accelerated molecular dynamics (GaMD) is an enhanced sampling that technique works by applying a harmonic boost potential to smooth biomolecular potential energy surface[27]. Since this boost potential exhibits a near Gaussian distribution, cumulant expansion to the second order ("Gaussian approximation") can be applied to achieve proper energetic reweighting[28]. GaMD allows for simultaneous unconstrained enhanced sampling and free energy calculations of large biomolecules[27]. GaMD has been successfully demonstrated on enhanced sampling of ligand binding, protein folding, protein conformational, as well as protein-membrane, protein-protein, and protein-nucleic acid interactions[29].

In 2020, Bhattarai et al.[30] combined complementary GaMD simulations and biochemical experiments to investigate mechanisms of the γ-secretase activation and the ε cleavage of wildtype (WT) and FAD-mutant APP substrates. GaMD simulations captured spontaneous activation of γ-secretase: First, the protonated D257 formed a hydrogen bond with the backbone carboxyl group of APP residue L49. Then, one water molecule was recruited between the two catalytic aspartates through hydrogen bonds. In this way, the water molecule was activated for nucleophilic attack on the carbonyl carbon of APP residue L49 to carry out the ε cleavage. GaMD simulations also revealed that APP FAD mutations I45F and T48P preferred ε cleavage at the L49–V50 amide bond, whereas M51F shifted the ε cleavage site to the T48–L49 amide bond, being highly consistent with experimental analyses of APP proteolytic products using mass spectrometry and western blotting[1,30]. Very recently, Pep-GaMD simulations were combined with further mass spectrometry and western blotting experiments to investigate tripeptide trimming of wildtype (WT) and FAD-mutant Aβ49 substrates by γ-secretase[31]. The Pep-GaMD simulations revealed remarkable structural rearrangements of both γ-secretase and Aβ49, where hydrogen-bonded catalytic aspartates and water were poised to carry out the ζ cleavage of Aβ49 to Aβ46. Furthermore, the tripeptide trimming required inclusion of endoproteolytic coproduct APP intracellular domain (AICD) with a positively charged N-terminus. The simulation findings were also highly consistent with biochemical experimental data[31,32].

In this work, we performed GaMD simulations and biochemical experiments in parallel to determine the effects of PS1 FAD mutations on γ-secretase activation for particularly the ε cleavage of APP. We selected six PS1 FAD mutations to investigate based on early age of disease onset and their representative locations relative to the transmembrane domains (TM) of PS1, including P117L (hydrophobic loop 1), I143T (TM2), L166P (TM3), L286V (TM6, active site), G384A (TM7, active site), and L435F (TM9) (Fig. 1a). Our GaMD simulations and biochemical experiments were largely consistent with each other and together provided important mechanistic insights into the effects of PS1 FAD mutations on structural dynamics and enzyme-substrate interactions of APP-bound γ-secretase.

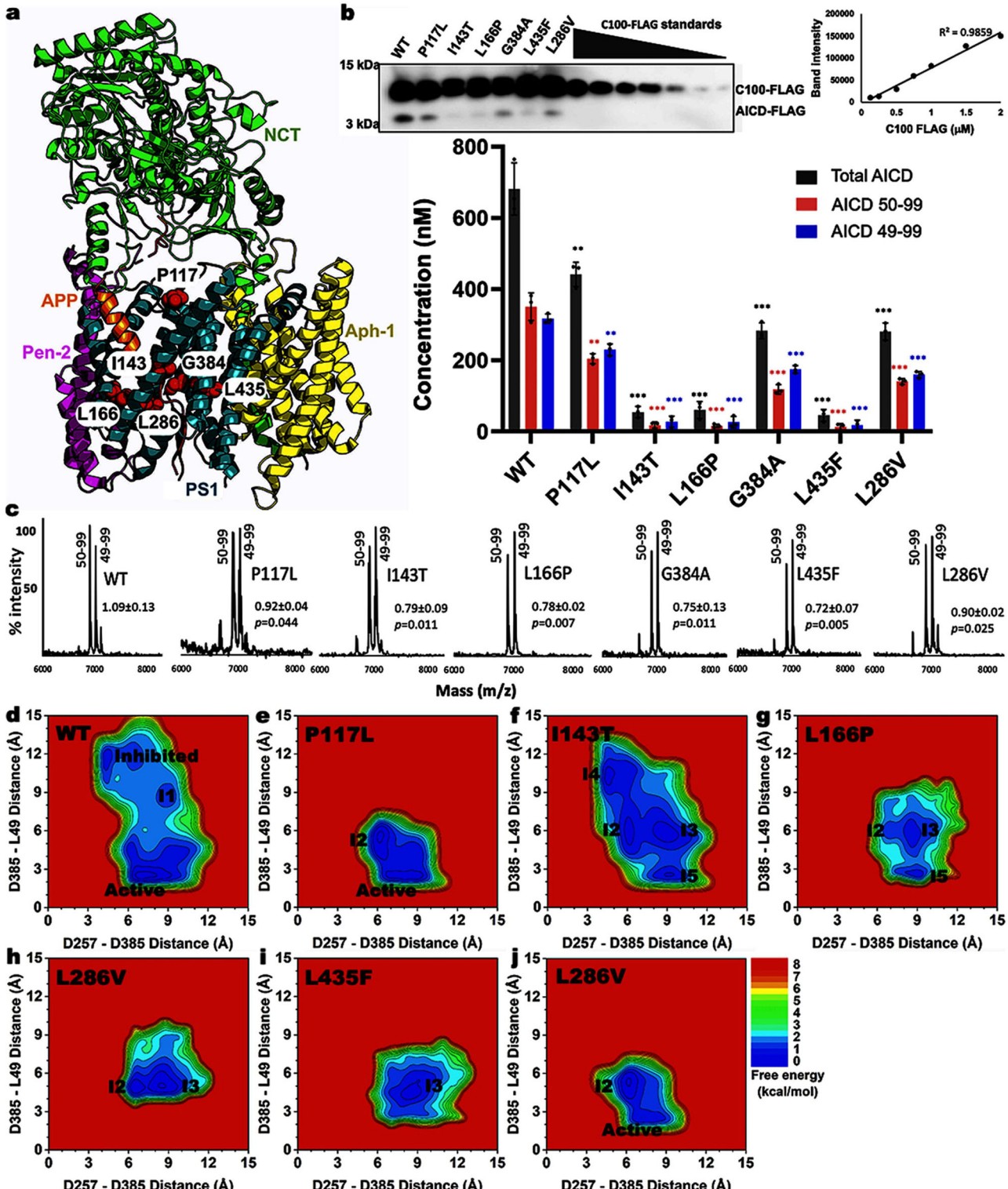

**Fig. 1 Summary of effects of PS1 FAD mutations on enzyme-substrate interactions of the APP bound γ-secretase complex. a** Cryo-EM structure of γ-secretase complex with APP bound (PDB: 6IYC) and locations of six PS1 FAD mutation residues (red spheres). The four components of γ-secretase are Nicastrin (NCT, green), Presenilin-1 (PS1, teal), Aph-1 (yellow) and Pen-2 (magenta). APP is shown in orange. **b** Anti-FLAG immunoblots and quantification of total AICD (black), AICD50-99 (red), and AICD49-99 (blue)-FLAG levels generated from the ε cleavage of APP by the WT and FAD mutants of γ-secretase by densitometry. Purified C100-FLAG at a range of known concentrations was used to generate a standard curve. **c** MALDI-TOF mass spectrometry (MS) detection of AICD50-99 and AICD49-99 products generated from the ε cleavage of APP by the WT and PS1 FAD mutants of γ-secretase. T-tests were performed, and the resulting p values were added along with the ratios to highlight the significance of the ratios determined for the PS1 FAD mutants. **d–j** 2D free energy profiles of the distance between PS1 residues D257 (atom Cγ) and D385 (atom Cγ) and distance between PS1 residue D385 (protonated oxygen) and APP residue L49 (carbonyl oxygen) in the WT (**d**) and P117L (**e**), I143T (**f**), L166P (**g**), G384A (**h**), L435F (**i**), and L286V (**j**) FAD mutants of APP bound γ-secretase. The low-energy conformational states are labeled "Active", "Inhibited", and "I1"–"I5".

## Results

**Cleavage of APP by WT and PS1 FAD-mutant γ-secretase in biochemical experiments.** To analyze the effects of six PS1 FAD mutations (P117L, I143T, L166P, G384A, L435F, and L286V) on the ε cleavage of APP by γ-secretase, cleavage assays using purified WT and FAD mutant γ-secretase were performed with purified, recombinant APP substrate C100-FLAG. Cleavage assay mixtures were subjected to quantitative western blotting using anti-FLAG primary antibodies. Known concentrations of C100-FLAG were run in parallel to make a calibration curve, where the band intensity was plotted versus the concentrations of FLAG-tagged C100, and a tight linear relationship was observed ($R^2 = 0.99$) (Fig. 1b). From this standard curve, the concentration of total AICD-FLAG products generated in the enzyme reaction mixtures were quantified. Quantification of the total AICD produced by FAD mutant γ-secretase revealed significantly decreased ε cleavage compared with WT γ-secretase (Fig. 1b). In particular, the concentration of AICD-FLAG produced by WT γ-secretase was ~686 ± 53 nM. This concentration decreased to ~474 ± 40 nM with P117L, ~284 ± 20 nM with L286V, ~274 ± 57 nM with G384A, ~90 ± 18 nM with L166P, ~78 ± 16 nM with I143T, and ~64 ± 17 nM with L435F PS1 FAD-mutant γ-secretase, respectively (Fig. 1b).

To further quantify the individual species of AICD, AICD generated in the cleavage assay were immunoprecipitated with anti-FLAG antibodies and monitored by matrix-assisted laser desorption/ionization time-of-flight (MALDI-TOF) mass spectrometry (MS). The ratios of signal intensities corresponding to AICD 49-99 to AICD 50-99 were calculated and this ratio along with total AICD quantified with western blotting was used to calculate the concentration of AICD 49-99 and AICD 50-99. The ratios between AICD50-99 and AICD49-99 were ~1.1 ± 0.1 with WT γ-secretase, ~0.9 ± 0.04 with P117L, ~0.9 ± 0.02 with L286V, ~0.8 ± 0.1 with I143T, ~0.8 ± 0.02 with L166P, ~0.8 ± 0.1 with G384A, and ~0.7 ± 0.1 with L435F PS1 FAD mutants, respectively, as detected by MALDI-TOF MS (Fig. 1c). Both species of AICD were significantly decreased for all the tested FAD mutants when compared to the WT γ-secretase (Fig. 1b). In particular, the concentration of AICD 50-99 flag decreased from ~363 ± 35 nM with WT γ-secretase to ~213 ± 15 nM with P117L, ~144 ± 8 nM with L286V, ~133 ± 8 nM with G384A, ~22 ± 12 nM with I143T, ~21 ± 12 nM with L166P, and ~17 ± 12 nM with L435F PS1 FAD mutant, respectively. The concentration of AICD 49-99 flag decreased from ~305 ± 28 nM with WT γ-secretase to ~222 ± 16 nM with P117L, ~157 ± 9 nM with L286V and G384A, ~26 ± 16 nM with L166P, ~26 ± 14 nM with I143T, and ~22 ± 16 nM with L435F PS1 FAD mutant, respectively (Fig. 1b). Since the average concentrations of AICD 50-99 and AICD 49-99 were relatively close, there was only very subtle shift in the ε cleavage of APP from the 49[th] to the 48[th] residue for all FAD mutants.

**Free energy profiles of the ε cleavage of APP by WT and PS1 FAD-mutant γ-secretase.** In parallel with biochemical experiments, all-atom dual-boost GaMD simulations were carried out on WT, P117L, I143T, L166P, G384A, L435F, and L286V PS1 FAD-mutant γ-secretase bound by APP (Supplementary Table 1). GaMD simulations recorded similar averages and standard deviations of the boost potentials among different systems, i.e., 13.5 ± 4.3 kcal/mol for the WT, 11.3 ± 4.0 kcal/mol for P117L, 14.0 ± 4.4 kcal/mol for I143T, 14.8 ± 4.5 kcal/mol for L166P, 13.8 ± 4.4 kcal/mol for G384A, 14.0 ± 4.4 kcal/mol for L435F, and 14.1 ± 4.1 kcal/mol for L286V PS1 FAD mutant γ-secretase, respectively (Supplementary Table 1). In this study, we chose to protonate D385 as its pKa value was calculated to be higher than

that of D257 (8.8 to 8.0, respectively) (Supplementary Table 2). In addition, the protonation of one catalytic aspartate (D385) allowed us to obtain comparable D257-D385 distances in our GaMD simulations with the available PDB structures of γ-secretase (Supplementary Table 3). In particular, distances between the Cγ atoms of catalytic aspartates D257-D385 calculated from GaMD simulations were 7.3 ± 1.9 Å for WT, 7.6 ± 1.1 Å for P117L, 8.2 ± 1.6 Å for I143T, 8.7 ± 1.0 Å for L166P, 8.1 ± 1.2 Å for G384A, 9.1 ± 1.2 Å for L435F, and 7.4 ± 1.0 Å for L286V PS1 FAD mutant γ-secretase (Supplementary Figs. 1–4). Meanwhile, the lowest D257-D385 distance could get to ~3.9 Å in the 5FN2[33] PDB structure, while most of the experimental D257-D385 and D257-A385 (in the 6IDF and 6IYC PDB) distances were between ~5 Å and ~9 Å[33–37]. The highest D257-D385 distance was ~11.5 Å, observed in the 5FN4[33] PDB structure (Supplementary Table 3). The ε cleavage of APP by γ-secretase can only be carried out when the two PS1 catalytic aspartates are at a suitable distance so that a nucleophilic water molecule can be recruited for the proteolytic reaction through water-bridged hydrogen bonding with the two aspartates[5,26,30]. Furthermore, the carbonyl group at the cleavage site on APP (residue L49) would form another hydrogen bond between the carbonyl oxygen and protonated carboxylic side chain of catalytic residue D385 in PS1 for proteolysis[5,26,30]. Therefore, the distance between the Cγ atoms of catalytic aspartates D257 and D385 in PS1 and the distance between PS1 residue D385 (protonated oxygen) and APP residue L49 (carbonyl oxygen) were calculated from the GaMD simulations and plotted in Supplementary Figs. 1–4. They were used as reaction coordinates to calculate two-dimensional (2D) potential mean force (PMF) free energy profiles to characterize the effects of PS1 FAD mutations on γ-secretase activation for ε cleavage of APP (Fig. 1). Overall, the WT γ-secretase sampled noticeably larger conformational space than the PS1 FAD mutants.

A total of seven different low-energy conformational states were identified from free energy profiles of the WT and six PS1 FAD mutants of γ-secretase bound by APP, namely "Active", "Inhibited", and five intermediate states "I1", "I2", "I3", "I4", and "I5" (Fig. 1). The "Active" state was observed in free energy profiles of the WT, P117L, and L286V PS1 FAD-mutant γ-secretase (Fig. 1d, e, j). In this state, the catalytic aspartates D257 and D385 in PS1 were ~7–9.5 Å apart and residue D385 formed a hydrogen bond with APP residue L49 at ~2.5–3 Å distance. At ~7–8 Å distance between the Cγ atoms, the two catalytic aspartates could recruit a water molecule through hydrogen bonds, poised for the ε cleavage of APP.

The "Inhibited" and "I1" low-energy conformational states were only observed in the free energy profile of WT γ-secretase (Fig. 1d). In the "Inhibited" state, the distance between catalytic aspartates D257 and D385 reduced to ~4 Å, whereas the distance between PS1 residue D385 and APP residue L49 increased to ~10–13 Å. In the "I1" state, the D257–D385 and D385–L49 distances became ~8–10 Å and ~7.5–9.5 Å, respectively.

The "I2" low-energy conformational state was observed in the free energy profiles of most of the PS1 FAD mutants, with the only exception of L435F (Fig. 1e–j). In this low-energy state, the distance between PS1 residues D257 and D385 decreased to ~6–7 Å, while the distance between PS1 residue D385 and APP residue L49 varies between PS1 FAD mutations in a range of ~5–7 Å.

The "I3" low-energy conformational state was identified from the free energy profiles of three PS1 FAD mutations, including I143T (Fig. 1f), L166P (Fig. 1g), and G384A (Fig. 1h). In this state, the distance between the two catalytic aspartates D257 and D385 ranged from ~8 to 10 Å, while the distance between D385 and L49 of APP was ~4 to 7 Å.

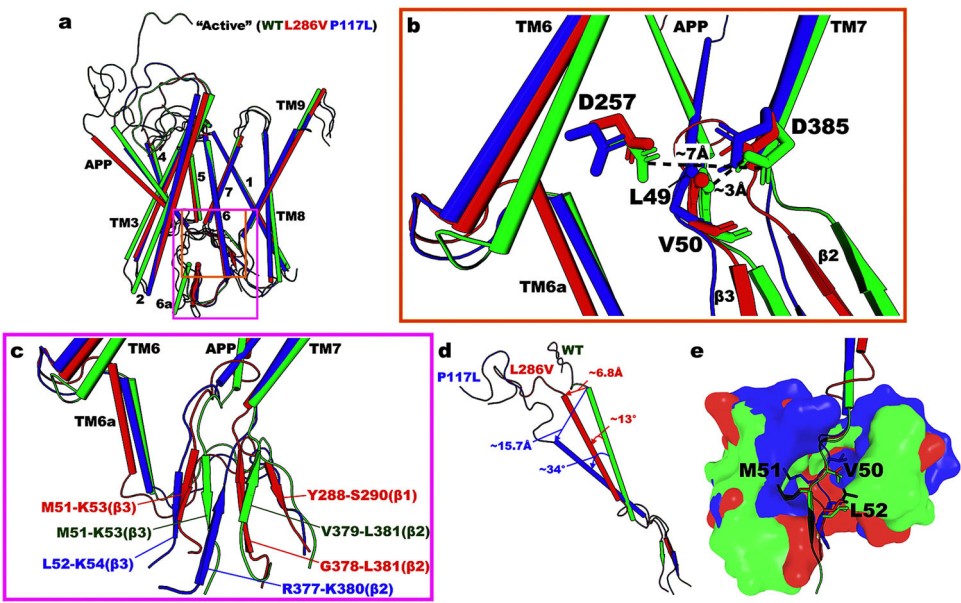

**Fig. 2 The "Active" low-energy conformational state in the WT, L286V, and P117L PS1 FAD mutants of APP-bound γ-secretase. a** The "Active" conformation of the APP-bound PS1 in WT (green), L286V (red), and P117L (blue) systems. **b** Active site of APP-bound PS1 in the "Active" WT, L286V, and P117L PS1. The distances between PS1 residues D257 and D385 are ~7.0 Å, ~7.0 Å, ~7.4 Å, and the distances between PS1 residue D385 and APP residue L49 are ~3.0 Å, ~2.9 Å, ~2.7 Å in the "Active" WT, L286V, and P117L γ-secretase, respectively. **c** Conformations of TM6a, TM7, and APP in the "Active" WT, L286V, and P117L PS1. **d** Conformations of the APP substrate in the "Active" WT, L286V, and P117L PS1. **e** Locations of APP substrate residues P1', P2', and P3' in the "Active" WT, L286V, and P117L PS1.

The "I4" low-energy state was only observed in the free energy profile of one PS1 FAD mutant, I143T (Fig. 1f). In this state, the distance between PS1 residues D257 and D385 was ~5–6 Å in the range between the "Inhibited" and "Active" states. However, the PS1 residue D385 and APP residue L49 was far apart, with a distance of ~10 to 11 Å.

The "I5" low-energy state was observed in the free energy profile of two PS1 FAD mutants, including I143T (Fig. 1f) and L166P (Fig. 1g). The distance between PS1 residues D257 and D385 centered around ~8.5–10 Å, while the distance between PS1 residue D385 and APP residue L49 ranged from ~2.5–4 Å in the "I5" state. The representative structures of all low-energy conformational states of APP-bound γ-secretase were provided in Supplementary Data 1.

**"Active" low-energy conformational state of γ-secretase bound by APP**. The "Active" low-energy conformational state was identified in the WT, P117L, and L286V γ-secretase (Fig. 1d, e, and j). This low-energy conformational state was characterized by the D257–D385 distance of ~7–9.5 Å and D385–L49 distance of ~2.5–3 Å. Representative PS1 and APP conformations of "Active" WT, P117L, and L286V γ-secretase obtained from structural clustering of their GaMD simulation snapshots using CPPTRAJ[38] were aligned for comparison in Fig. 2. The $C_\alpha$-RMSD of PS1 and APP of "Active" L286V and P117L relative to WT were ~1.7 and ~1.7 Å, respectively, illustrating the similarity between these conformations. However, it is worth noting that the intracellular ends of TM2, TM3, TM6a, and TM8 moved inwards in the L286V and P117L PS1 mutants compared to WT γ-secretase (Fig. 2a).

The active site in the WT, P117L, and L286V PS1 were compared in Fig. 2b. Overall, PS1 residues D257 and D385 and APP residue L49 were well aligned among the three simulation systems. The distances between the Cγ atoms of residues D257 and D385 in WT, L286V, and P117L PS1 were ~7.0, ~7.0, and ~7.4 Å, respectively. These distances were all suitable for the

catalytic aspartates to activate nucleophilic water to carry out the proteolytic reaction. Notably, the side chains of D257 and D385 could rotate in the simulation systems (Fig. 2b). The protonated oxygen of D385 formed a hydrogen bond with the backbone carbonyl oxygen of APP residue L49 in the WT, L286V, and P117L PS1 systems with distances of ~3.0, ~2.9, ~2.7 Å, respectively.

Next, we examined the secondary structures of the PS1 and substrate near the active site in Fig. 2c as they appeared different across the three systems. In the "Active" WT low-energy conformational state, while the β1 domain (connected to TM6a) remained unstructured, the β2 strand (connected to TM7) formed a hybrid β-sheet with the C-terminal β3 strand of APP, between PS1 residues V379–L381 and APP residues M51–K53. In the "Active" L286V low-energy conformational state, antiparallel β-strands were formed between the β1, β2, and β3 domains, involving PS1 residues Y288–S290 and G378–L381 and APP residues M51–K53. In the "Active" low state of P117L PS1 FAD mutant, the antiparallel β-strands were formed between the β2 domain and β3 APP C-terminus, involving PS1 residues R377–K380 and APP residues L52–K54.

The helical domain of APP tilted in the P117L and L286V PS1 by ~9 and 34 degrees, respectively, compared to that in WT PS1 (Fig. 2d). Meanwhile, the extracellular end of the APP helical domain in the L286V and P117L PS1 FAD mutants moved by ~6.8 and ~15.7 Å, respectively. The length of the APP helical domain also decreased from ~28.1 Å in the WT PS1 to ~24.5 Å and ~22.5 Å in the L286V and P117L mutants, respectively.

The locations of P1', P2', and P3' residues and corresponding S1', S2', and S3' subpockets in the "Active" WT, L286V, and P117L PS1 FAD-mutant γ-secretase were compared in Fig. 2e. Here, P1', P2', and P3' referred to APP residues that were one, two, and three residues away downwards, respectively, from the APP cleavage side residue L49 (i.e., V50, M51, and L52). The corresponding S1', S2', and S3' subpockets consisted of residues that were within 5 Å of APP substrate residues P1' V50, P2' M51,

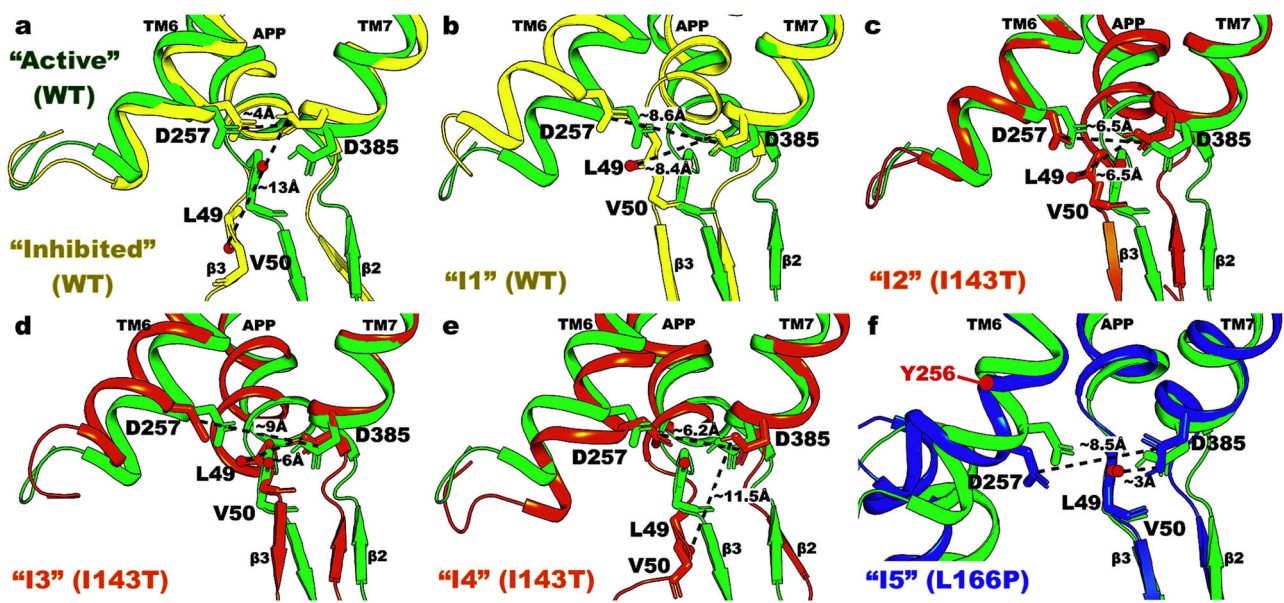

**Fig. 3 Distinct low-energy conformational states of catalytic aspartates and enzyme-substrate interactions at the active site of WT and PS1 FAD mutant γ-secretase compared to "Active" WT conformation. a** The "Inhibited" state, for which the distance between PS1 residues D257 and D385 is ~4.1 Å, and the distance between PS1 residue D385 and APP residue L49 is ~12.5 Å. **b** The "I1" state, for which the distance between PS1 residues D257 and D385 is ~8.6 Å, and the distance between PS1 residue D385 and APP residue L49 is ~8.4 Å. **c** The "I2" state, for which the distance between PS1 residues D257 and D385 is ~6.5 Å, and the distance between PS1 residue D385 and APP residue L49 is ~6.5 Å. **d** The "I3" state, for which the distance between PS1 residues D257 and D385 is ~8.8 Å, and the distance between PS1 residue D385 and APP residue L49 is ~6.1 Å. **e** The "I4" state, for which the distance between PS1 residues D257 and D385 is ~6.2 Å, and the distance between PS1 residue D385 and APP residue L49 is ~11.5 Å. **f** The "I5" state, for which the distance between PS1 residues D257 and D385 is ~8.5 Å, and the distance between PS1 residue D385 and APP residue L49 is ~2.9 Å. The "Active" WT low-energy conformation is shown in green for reference.

and P3' L52. The RMSD of the $C_\alpha$ atoms in the P1', P2', and P3' residues of APP in the L286V PS1 mutant was ~0.1 Å relative to that in the WT PS1. On the other hand, RMSD of the $C_\alpha$ atoms in the P1', P2', and P3' residues of APP in the P117L PS1 mutant increased to ~0.2 Å. In addition, RMSD of the $C_\alpha$ atoms in the corresponding S1', S2', and S3' subpockets relative to WT PS1 was lower in the L286V than in the P117L mutant, with respective values of ~1.6 Å compared to ~3.7 Å. The full lists of residues constituting the S1', S2', and S3' subpockets in the three systems can be found in Supplementary Table 4. It is worth noting that the total numbers of residues constituting the S1', S2', and S3' subpockets in the L286V and P117L PS1 mutants were both 36 and larger than that in the WT PS1, which was 23 (Supplementary Table 4).

**Intermediate low-energy conformational states of γ-secretase bound by APP.** Besides the "Active" state, six other intermediate low-energy conformational states were identified from the free energy profiles of WT and PS1 FAD-mutant γ-secretase, including "Inhibited", "I1", "I2", "I3", "I4", and "I5". Representative PS1 and APP conformations of the intermediate low-energy states were compared to the "Active" state of WT PS1 in Figs. 3–4 and Supplementary Figs. 5–8.

Different active site conformations in the intermediate low-energy states were compared to the "Active" state of WT PS1 in Fig. 3. In the "Inhibited" low-energy state, the distance between the two catalytic aspartates D257 and D385 in PS1 decreased from ~7.0 Å to ~4.1 Å, whereas the distance between residue D385 and APP residue L49 increased from ~3.0 Å to ~12.5 Å (Fig. 3a). The two catalytic aspartates moved towards each other, resulting in the formation of a hydrogen bond between the proton of D385 and carbonyl oxygen of D257. Meanwhile, residue L49 in APP moved downwards by ~6 Å, providing room for the

formation of D257–D385 hydrogen bond (Fig. 3a). In the "I1" low-energy state, the PS1 TM6 and APP β3 strand moved away from PS1 TM7, which increased the D257–D385 and D385–L49 distances to ~8.6 and ~8.4 Å, respectively (Fig. 3b). The "I2" state was similar to "I1", except that TM7 moved inwards relative to the "Active" state in WT PS1, reducing both the D257–D385 and D385–L49 distances to ~6.5 Å (Fig. 3c). In the "I3" state, TM6 moved slightly outwards and APP substrate moved slightly upwards relative to the "Active" WT conformation. These movements increased the distance between PS1 residues D257 and D385 to ~9 Å and reduced the distance between PS1 residue D385 and APP residue L49 to ~6 Å (Fig. 3d). The "I4" was the only intermediate conformational state where TM6 shifted inwards relative to the "Active" WT, reducing the D257–D385 distance to ~6.2 Å. In addition, APP residue L49 moved downwards for ~6 Å, increasing the D385–L49 distance to ~11.5 Å (Fig. 3e). Notably, the antiparallel β strands between the β2 domain near PS1 TM7 and the β3 domain in the APP C-terminus were maintained in all but two of the intermediate low-energy states (i.e., "Inhibited" and "I4"). Furthermore, the backbone carbonyl group of APP residue L49 pointed towards D257 instead of the protonated D385 in three of the intermediate states ("I1", "I2", and "I3").

In the "I5" low-energy conformational state, the protonated oxygen atom of D385 in PS1 formed a hydrogen bond with APP residue L49 at a ~2.9 Å distance. However, the distance between PS1 residues D257 and D385 increased to ~8.5 Å due to a helical stretch around residue Y256 in TM6 (Fig. 3f). The stretch moved residue D257 downwards relative to the "Active" WT and increased the D257–D385 distance out of the ~7–8 Å range required for activation of γ-secretase. In fact, with ~7 Å distance between D257–D385, the active site in the "Active" WT conformational state was properly poised for the two catalytic aspartates to recruit a water molecule. The water molecule was

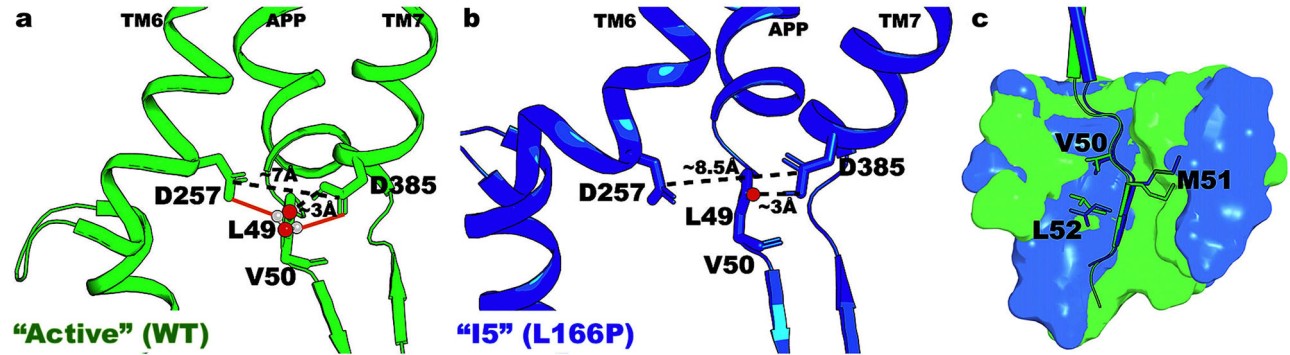

**Fig. 4 Comparison between the "Active" and "I5" low-energy conformational states in γ-secretase. a** The "Active" low-energy conformational state of WT PS1, where the distance between residues D257 and D385 is ~7.0 Å. A water molecule formed hydrogen bonds with the two catalytic aspartates and poised for the ε cleavage of the amide bond between residues L49–V50 of APP. **b** The "I5" low-energy conformational state of L166P FAD mutant PS1, where the distance between the two catalytic aspartates D257 and D385 is too large at ~8.5 Å to trap a water molecule for the ε cleavage of APP. **c** Location of APP substrate residues P1′, P2′, and P3′ in the "I5" low-energy conformational state compared to "Active" WT. The "Active" and "I5" low-energy conformational states are shown in green and blue, respectively.

made nucleophilic and properly oriented to carry out the ε cleavage of APP residue L49 through the hydrogen bonds formed with the carboxylic side chains of residues D257 and D385 (Fig. 4a). To further examine the water dynamics during γ-secretase activation for ε cleavage of APP, we reproduced a 100 ns GaMD simulation of the "Active" WT γ-secretase, starting from the 1200 ns checkpoint of Sim1, and saved the coordinates of not only proteins and substrates but also lipids, ions, and water molecules (Supplementary Fig. 9). The time courses of the D257-D385 and D385-L49 distances were calculated and shown in Supplementary Fig. 9a. Upon the formation of the D385-L49 hydrogen bond at ~3 Å distance while the PS1 residues D257 and D385 maintained ~6–8 Å distance, a water molecule was recruited (Supplementary Fig. 9b) and trapped between the two catalytic aspartates (Supplementary Fig. 9c) to carry out the proteolytic reaction in the "Active" conformation. This has also been observed in our previous study[30]. At the D257–D385 distance of ~8.5 Å in the "I5" state, the active site was so "open" that no water molecule could be properly stabilized between the catalytic aspartates for the proteolytic reaction (Fig. 4b). Furthermore, the locations of P1′, P2′, and P3′ residues and corresponding S1′, S2′, and S3′ subpockets were compared between the "Active" and "I5" low-energy conformational states (Fig. 4c). Here, the $C_\alpha$-RMSD of P1′, P2′, and P3′ residues of APP in the "I5" low-energy conformation relative to "Active" WT was ~0.24 Å, and the $C_\alpha$-RMSD of S1′, S2′, and S3′ subpockets was ~0.91 Å. The total number of residues constituting the S1′, S2′, and S3′ subpockets of the "I5" low-energy conformational state (24) was similar to that of "Active" WT conformation (23) (Supplementary Table 4).

We compared the entire PS1 subunit bound to APP in the intermediate low-energy states to the "Active" WT state in Supplementary Figs. 5–8. A number of notable differences were identified in the APP substrate (Supplementary Fig. 6), the β1, β2, and β3 domains (Supplementary Fig. 7), and PS1 TM8 (Supplementary Fig. 8). First, the APP helical domain tilted in all the intermediate conformations relative to the "Active" WT conformation, with the largest tilts observed in the "Inhibited" and "I1", and the smallest tilt in the "I5" state (Supplementary Fig. 6). Compared to the "Active" WT conformation, the extracellular end of APP moved by ~11.7 Å in the "Inhibited", ~11.8 Å in the "I1", ~9.3 Å in the "I2", ~10.2 Å in the "I3", ~11.2 Å in the "I4", and ~6.9 Å in the "I5", with respective tilt angles of ~24°, ~25°, ~17°, ~14°, ~16°, and ~14° (Supplementary Fig. 6). The length of APP helical domain also changed ~28.1 Å in

the "Active" WT state to different values in the intermediate conformations. It decreased to ~27.3 Å in the "Inhibited", ~25.8 Å in the "I1", and ~6.9 Å in the "I4", while increased to ~30.7 Å in the "I2", ~30.1 Å in the "I3", and ~29.0 Å in the "I5".

Second, the β1, β2, and β3 domains (connected to TM6a, TM7, and APP, respectively) also varied in their conformations in the intermediate low-energy conformational states relative to the "Active" WT conformation (Supplementary Fig. 7). In the "Inhibited" and "I4" states, the β3 domain lost its β-strand secondary structure as it moved away from β2, while the β2 formed anti-parallel β-strands with β1. (Supplementary Fig. 7a, e). In the "I1", "I3", and "I5" states, the β1, β2, and β3 domains formed antiparallel β-sheets with one another (Supplementary Fig. 7b, d, f). Notably, the β2 and β3 strands extended in the "I1" state, involving residues R377–G378 near TM7 and K54 of APP. In the "I2" state, the secondary structures of the β domains were similar to those in the "Active" WT conformation (Supplementary Fig. 7c). Furthermore, TM6a tilted noticeably in the "I1", "I2", and "I3" states compared to "Active" WT, with the largest tilt observed in the "I3" conformation (Supplementary Fig. 7b, c, d).

Third, the intracellular end of TM8, which lies at the interface of PS1 and APH-1 subunits, all moved away from the PS1 TM bundle towards the APH-1 subunit in the intermediate low-energy conformations (Supplementary Figs. 5, 8). Relative to the "Active" WT conformation, the TM8 intracellular end moved by ~7.5 Å in the "Inhibited", ~5.9 Å in the "I1", ~7.0 Å in the "I2", ~7.4 Å in the "I3", ~7.7 Å in the "I4", and ~5.0 Å in the "I5" state. In addition, the helical domain of TM8 in the "I2" conformation was distorted at residue L423 (Supplementary Fig. 8c).

**Secondary structures of APP substrate in WT and PS1 FAD-mutant γ-secretase.** Representative time courses of APP secondary structures for the WT and PS1 FAD-mutant γ-secretase were shown in Fig. 5, while time courses of APP secondary structure from the remaining GaMD simulations were plotted in Supplementary Figs. 10–13. Overall, APP secondary structures in WT PS1 changed notably to those in PS1 FAD mutants, even for those whose proteolytic activity reduced only slightly, such as P117L. In the WT γ-secretase, residues K28–V46 were mostly helical, with few fluctuations to become 3-10-helices at residues A42–V46, in the representative Sim1, where the "Active" conformation was observed (Fig. 5a). Notably, for ~50 ns between 350 and 400 ns, residues V44-I45 turned 3-10-helical for the first half. The APP C-terminus was extended β-sheets during parts of

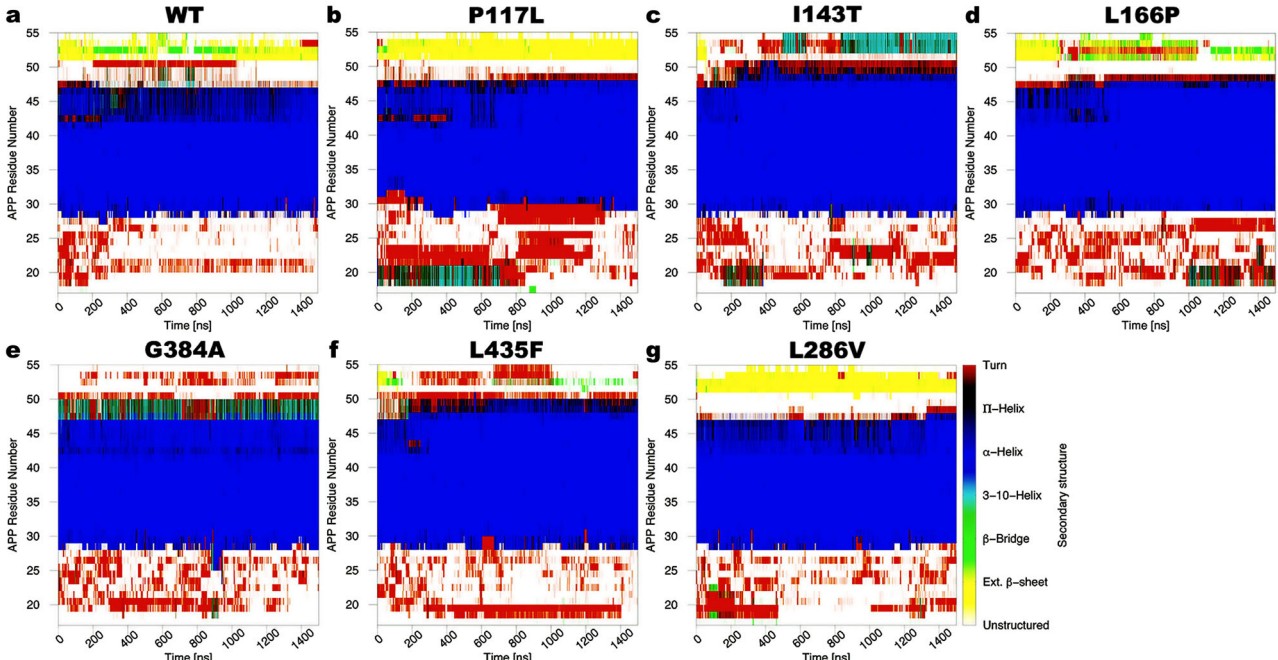

**Fig. 5 Time-dependent secondary structures of APP bound to γ-secretase calculated from the GaMD simulations.** Time courses of the APP secondary structures in the (**a**) WT (Sim1), (**b**) P117L (Sim1), (**c**) I143T (Sim1), (**d**) L166P (Sim2), (**e**) G384A (Sim3), (**f**) L435F (Sim1) and (**g**) L286V (Sim1) PS1 FAD mutant γ-secretase calculated from representative GaMD simulations. Results from other simulations are plotted in Supplementary Figs. 10–13.

Sim1. In particular, residues I47–L50 incidentally turned 3-10-helices, whereas residues M51–K53 were mostly extended β-sheets in Sim1 (Fig. 5a). The time courses of APP secondary structures in Sim2 and Sim3 of WT γ-secretase were shown in Supplementary Fig. 10.

In the P117L PS1 FAD mutant γ-secretase, APP secondary structures were similar between the representative (Sim1) (Fig. 5b) and other simulations (Supplementary Fig. 11a, b). First, the N-terminus of APP, involving residues V18–G29, could be helical. Residues V18–E22, specifically, adopted 3-10-helices in Sim1 (Fig. 5b) or α-helix conformation during Sim2 and Sim3 (Supplementary Fig. 11a, b). The length of APP helical domain in the P117L FAD mutant remained similar to that in the WT PS1, covering residues A30–V46. However, the APP C-terminus was β-strand in residues M51–K54 (Fig. 5b and Supplementary Fig. 11).

In the I143T PS1 mutant, the representative time course of APP secondary structures (Sim1) showed a slight increase in the helical length involving residues K28–L49 compared to K28–V46 of APP in the WT PS1 (Fig. 5c). Compared to other systems, APP residue A42 was solely α-helical in this mutant, while residue I47 could be either helical or turned (Fig. 5c and Supplementary Fig. 12). Furthermore, only in Sim2 were residues M51–K53 observed as extended β-sheet for most of the simulation (Supplementary Fig. 12c). In Sim1 and Sim3, this portion of APP C-terminus occasionally became 3-10-helices between residues L52–K54 observed during ~450–610 ns and ~820–1200 ns of Sim1 (Fig. 5c).

In the L166P PS1 mutant, the average APP helical length included residues K28–I47 (Fig. 5d and Supplementary Fig. 13a, b). Furthermore, residues T43–V45 could be 3-10-helices and turns. Here, residues L17–N27 at the N-terminus of APP were mostly unstructured or turns, with some fluctuations to 3-10-helices, while residues M51–K53 at the C-terminus of APP could be mostly extended β-sheets (Fig. 5d and Supplementary Fig. 13).

The secondary structures of APP in the G384A PS1 mutant were mostly similar to other simulation systems (Fig. 5e and Supplementary Fig. 12). However, two notable differences could be identified from the simulation time courses. First, residues M51–K53 in the APP C-terminus were mostly turns or unstructured across all three simulations. Second, residues I47–L49 mostly adopted the 3-10-helical conformation, unlike other simulation systems where α-helix were the preferred conformations for this region (Fig. 5e and Supplementary Fig. 12).

For the remaining PS1 FAD mutants, including L435F (Fig. 5f) and L286V (Fig. 5g), the APP secondary structures were almost identical to those in certain PS1 FAD mutants as described above. In particular, the time courses of the L435F FAD mutant (Fig. 5f and Supplementary Fig. 13c–d) were similar to those of L166P (Supplementary Fig. 13a) and G384A (Fig. 5e and Supplementary Fig. 12c–d) PS1 FAD mutants. For L286V, the secondary structures of APP were comparable to those in the P117L PS1 mutant, being consistent with the high similarity between the free energy profiles of these two systems (Figs. 1, 5, and Supplementary Fig. 11).

## Discussion

In this work, we have presented the first dynamic models for cleavage of amyloid precursor protein (APP) by PS1 FAD mutants of γ-secretase, which were consistent with mass spectrometry (MS) and western blotting biochemical experiments. Through the quantifications of the total AICD species produced by WT and PS1 FAD mutant γ-secretase, our biochemical experiments revealed significantly decreased ε-cleavages of APP by the PS1 FAD mutants compared to WT γ-secretase[39–41]. Since the PS1 FAD mutants mostly reduced ε-cleavage efficiency, the catalytic efficiency should be reduced, which means lower values of $k_{cat}/K_M$. The reason the experimental results specifically show reduction in $k_{cat}$ is that they are performed under conditions of substrate saturation. Under these conditions the rate is only determined by $k_{cat}$ and the concentration of enzyme, the latter which is kept constant. Therefore, a reduced rate of AICD product formation is due to a corresponding decrease in the $k_{cat}$.

GaMD simulations were carried out in parallel to explain the biochemical results in atomistic details. From the 2D free profiles calculated from GaMD simulations, important low-energy conformational states were identified for each simulation system of γ-secretase. The free energy landscapes and low-energy conformational states were explored in detail, which allowed us to deduce the effects of PS1 FAD mutants on the proteolytic activity of γ-secretase. Here, our main conclusion was that the PS1 FAD mutant γ-secretase stabilized the active sites of the enzyme-substrate complexes, which was distinctly different from previous studies, which suggested that PS1 FAD mutants destabilized the enzyme-substrate complexes, causing the earlier releases of longer Aβ peptides[10,22,42–45].

Our experimental method has already been validated in one other recent study[15]. In that study, we quantified all proteolytic events by γ-secretase on C100-Flag substrate with WT and 14 FAD-mutant substrates. For these 15 variants of C100-Flag, the quantification of AICD-Flag using the western blotting method (with C100-Flag itself used as the standard) gave results that were highly consistent with those from LC-MS/MS quantification of small peptide carboxypeptidase coproducts[15]. In deducing the production of all Aβ variants from these data, we found that total AICD equaled total Aβ in all cases. Moreover, the sums of Aβ peptides produced along the Aβ40-producing pathway from Aβ49 and along the Aβ42-producing pathway from Aβ48 were equivalent to their corresponding AICD products (AICD50-99 and AICD49-99, respectively)[15]. If the quantification of AICD-Flag using C100-Flag as the standard were inaccurate, such close agreement between AICD and Aβ products would not have been observed. Moreover, while the AICD bands produced from I143T, L166P and L435F were extremely faint, they were visible and within range of the standard curve (stronger than the band of the lowest concentration standard) (Fig. 1b).

The experimental effects seen on AICD production with the specific PS1 mutations under study here have also been reported by other groups[13,17,46–48]. According to Chávez-Gutiérrez et al., AICD production was reduced by the G384A, L166P, and I143T PS1 FAD mutants[49]. Severely compromised γ-secretase activity with the L435F PS1 FAD mutant has been previously reported by several groups[17,46–48]. For the L286V PS1 FAD mutant, we are only aware of our own previous report on its effect on ε cleavage to AICD[13]. In that report, we did not see decreased AICD production vis-à-vis WT enzyme; however, γ-secretase components were overexpressed in Chinese hamster ovary (CHO) cells with endogenous enzyme present, and assays were conducted using isolated membranes, not purified enzyme complexes. Therefore, we favor the results from our current study, which were obtained with purified enzyme and more rigorous quantification of AICD using a standard curve. For the P117L PS1 FAD mutant, we are unaware of any reports on the overall proteolytic activity, only Aβ42/40 ratios.

We performed four additional 1.5μs cMD simulations on each of four representative APP-bound γ-secretase systems, including the WT and the P117L, I143T, and L166P PS1 FAD mutants. The time courses of the D257-D385, D385-V50, D385-L49, and D385-T48 distances calculated from the cMD simulations were plotted in Supplementary Fig. 14. 2D free energy profiles of the (D257-D385, D385-L49), (D257-D385, D385-V50), or (D257-D385, D385-T48) distances (Supplementary Fig. 15) were calculated and compared with those from GaMD simulations (Fig. 1 and Supplementary Fig. 16). For both the cMD and GaMD simulations, the low-energy conformational states calculated from both the D385-V50 and D385-T48 distances matched those calculated from the D385-L49 distances. Moreover, GaMD sampled larger conformational space than the cMD simulations and uncovered additional low-energy conformational states in the WT, I143T,

and L166P FAD mutant γ-secretase systems (Fig. 1 and Supplementary Fig. 16). In particular, the WT, I143T, and L166P simulation systems visited two ("Inhibited" and "I1"), one ("I4"), and one ("I3") additional low-energy conformational states in the GaMD simulations than in the cMD simulations, respectively (Fig. 1d, f, g and Supplementary Fig. 15a, c, d). In the P117L simulation system, both GaMD and cMD uncovered two low-energy conformational states, i.e., the "Active" and "I2" (Fig. 1e and Supplementary Fig. 15b). These findings demonstrated the enhanced sampling power of GaMD in simulations of large biomolecules such as γ-secretase.

GaMD simulations of WT γ-secretase for ε cleavage of APP led to three primary low-energy conformational states, including "Inhibited", "I1", and "Active" (Fig. 1d). In the "Inhibited" low-energy conformation, the two catalytic aspartates D257 and D385 formed a hydrogen bond with each other[50], precluding their interaction with and activation of a water molecule, while APP residue L49 was located downstream and far away (Fig. 3a). In the "I1" low-energy conformational state, the active site opened up as residues D257 and D385 moved away from one another, while APP residue L49 moved upwards compared to the "Inhibited" low-energy conformation (Fig. 3b). As the APP substrate was properly located inside the active site, its β3 strand (involving APP residues M51–K53) was formed through the hydrogen bonds with the β2 strand connected to PS1 TM7 (involving PS1 residues V379–L381)[50]. This finding was highly consistent with previous simulation studies, in which the repeated formations of β-strands in several solvent-exposed regions of presenilin were observed[22,43,51,52]. Afterwards, the catalytic aspartates D257 and D385 drew closer to each other, at a ~7–8 Å distance in the "Active" conformation, to recruit a water molecule poised for the proteolytic reaction (Fig. 4a). The water molecule was made nucleophilic and properly oriented through the hydrogen bonds with the carboxylic side chains of D257 and D385, while the backbone carbonyl of APP residue L49 was made more electrophilic through a hydrogen bond formed with the protonated oxygen atom of residue D385 (Figs. 2b, 4a). With all the proper conditions met, γ-secretase activation for ε cleavage of APP was carried out in the "Active" low-energy conformational state (Fig. 6b). This finding agrees well with our previous study[30] even though a different aspartate in PS1 (D385) was protonated because of the higher pKa value calculated by PROPKA3[53,54]. Furthermore, given the locations of the low-energy conformational states in the WT free energy profile (Fig. 1d), it was plausible that transitions could take place between the "Inhibited" and "I1" as well as "I1" and "Active" conformations.

The effects of PS1 FAD mutants on γ-secretase activation for ε cleavage of APP could be deduced from the respective 2D free energy profiles, low-energy conformational states associated with each mutant, and changes in the APP substrate. As described in the Results section, the conformational space of WT γ-secretase (Fig. 1d) was noticeably larger compared to the PS1 FAD mutants, especially in the D385–L49 distance. In particular, the distance between PS1 residue D385 and APP residue L49 could range from ~2 to ~15 Å, and the distance between PS1 residues D257 and D385 was between ~3 and ~13 Å in WT γ-secretase. When we compared the "Active" low-energy conformations among WT, L286V, and P117L PS1 FAD mutants, TM2, TM3, TM6a, and TM8 all moved inwards in the two FAD mutants compared to WT γ-secretase (Fig. 2a). Therefore, the active site of WT γ-secretase appeared more flexible than the PS1 FAD mutants. This was further reinforced by the finding that fewer PS1 residues constituted the S1', S2', and S3' subpockets of the "Active" WT compared to "Active" P117L and L286V (23 vs. 36 and 36) (Supplementary Table 4) (i.e., the FAD-mutant enzymes had more contact with the corresponding APP residues P1', P2',

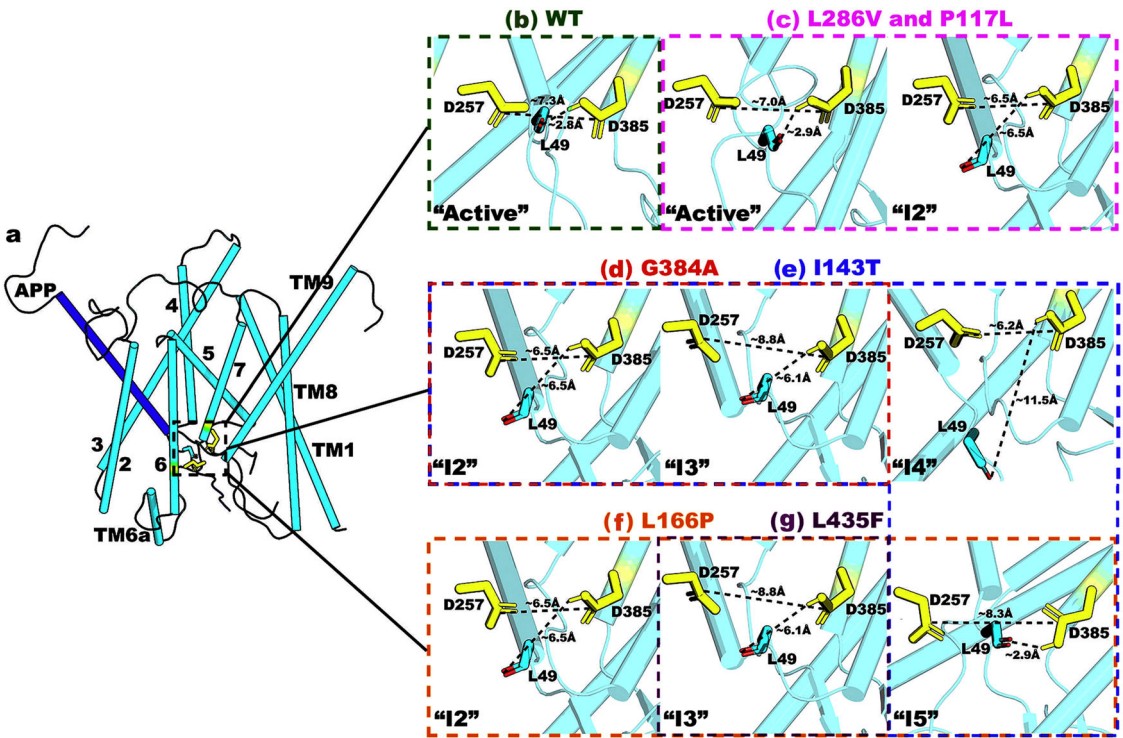

**Fig. 6 Summary of the effects of PS1 FAD mutations on the ε cleavage of APP by γ-secretase. a** Structural model of APP-bound PS1. The APP substrates are more tilted in the PS1 FAD mutants compared to WT γ-secretase. **b** The "Active" WT γ-secretase. **c** The active site of the L286V and P117L PS1 FAD mutants. **d**, **e** The active site of G384A ("I2"–"I3") and I143T ("I2"–"I5") PS1 FAD mutants. **f**, **g** The active site of the L435F ("I3") and L166P ("I2"–"I3" and "I5") PS1 FAD mutants.

and P3'). Furthermore, the APP helical domain tilted less in the "Active" WT state than in the other low-energy conformations, including the "Active" P117L and "Active" L286V (Fig. 2d and Supplementary Fig. 6). Nevertheless, it was worth noting that the β3 strand of APP was formed in all three "Active" low-energy conformations (WT, P117L, and L286V), being consistent with previous studies[22,43,51,52].

We showed that flexibility of the active site played an important role in γ-secretase activation for ε cleavage of APP. Even for PS1 FAD mutants such as P117L and L286V where the "Active" low-energy conformational state was identified, the conformational space of the active site in PS1 shrunk noticeably with respect to both D257–D385 and D385–L49 distances relative to WT γ-secretase. In particular, the distance range between PS1 residues D257 and D385 decreased to ~5–11 Å and ~3–11 Å in the P117L and L286 PS1 mutants, respectively, while the range for D385–L49 distance shrunk to ~2–7 Å in both FAD mutants (Fig. 1e, j). These two PS1 FAD mutants sampled only two stable low-energy conformational states, including "Active" and "I2" (Fig. 6c). Even in their respective "Active" states, the active site in PS1 and bound APP substrate were restricted, evidenced by the total number of interacting residues constituting the S1', S2', and S3' subpockets. In addition, the P117L and L286V PS1 mutants sampled the "I2" state, in which the active site appeared "semi-closed", with the two catalytic aspartates moving close to each other (Figs. 1e, j, 6c). Here, a "semi-closed" active site is defined as having a ~D257–D385 distance between ~6 and ~6.5 Å[26]. Furthermore, the free energy landscape near "I2" in the L286V PS1 FAD mutant complex could extend to ~4 Å D257–D385 distance (Fig. 1j). A distance of ~4 Å between D257 – D385 signified a closed active site, in which a hydrogen bond was formed between the two catalytic aspartates (as in the "Inhibited" low-energy conformation and 5FN2[33] PDB structure) (Fig. 3a). This

observation supported our experimental finding that L286V showed a lower proteolytic activity compared to P117L (Fig. 1b), as it was more effective in closing the active site to APP.

The I143T PS1 FAD mutant sampled four intermediate low-energy conformational states in its free energy profile, including the "I2", "I3", "I4", and "I5" (Fig. 1f). In the "I2" and "I4" states, the distance between two catalytic aspartates remained at ~6 Å, while the D385–L49 distance could be either ~6–7 Å in "I2" or ~10–11 Å in "I4" (Fig. 1f). The presence of these two conformations in its free energy profile indicated that I143T had the ability to "semi-close" the PS1 active site, preventing the APP substrate from being properly located for its ε cleavage. In the "I3" and "I5" states, the distance between PS1 residues D257 and D385 stayed at ~8.5–10 Å, while the D385–L49 distance could be either ~5–7 Å in the "I3" and ~2–4 Å in the "I5". As described in the Results section (Fig. 4b), a hydrogen bond could be formed between the protonated oxygen atom of D385 and carbonyl group of L49, but the two catalytic aspartates were too far apart to recruit a water molecule to carry out the ε cleavage. As such, this FAD mutant appeared to disrupt the D257 and D385 distance. Therefore, the I143T PS1 FAD mutant could either prevent the APP substrate from aligning within the active site (illustrated in "I2", "I3", and "I4" states) or disrupt the catalytic aspartate distances (shown in "I3" and "I5) (Fig. 6e).

The free energy landscapes of the active subpocket in the remaining PS1 FAD mutants, including L166P (Fig. 1g), G384A (Fig. 1h), and L435F (Fig. 1i), all shrunk noticeably compared to WT γ-secretase, showing that the APP-bound active site in PS1 was more restricted in these three FAD mutants. Three intermediate low-energy conformational states were identified from the free energy profiles of the L166P PS1 FAD mutant, including "I2", "I3", and "I5". As described above, the presence of "I2" and "I3" states suggested that the FAD mutant prevented the APP

substrate from entering the active site, while the presence of "I3" and "I5" states suggested that this mutant increased the D257–D385 distance. However, given the relative lower free energy of "I5" compared to "I2" and "I3" (Fig. 1g), the primary effect of the L166P FAD mutant appeared to be disrupting the D257 and D385 distance (Figs. 1g, 6g). The primary effect of the L435F PS1 FAD mutant was similar to that of the L166P as its free energy profile sampled mostly the "I3" state, which extended towards the "I5" state (Fig. 1i). This was to be expected as residue L435 in PS1 is located between the two catalytic aspartates D257 and D385. Its mutation to a larger residue such as phenylalanine could create steric clashes within the PS1 active site, thereby increasing the D257–D385 distance[26] (Fig. 6f). This finding was consistent with that by Chen and Zacharias[26], even though their simulations were performed on apo γ-secretase. Chen and Zacharias found that mutation of L435, which was located in close proximity to the active site, to phenylalanine shifted the D257-D385 Cγ-distance to larger distances and increased the equilibrium Cγ-Cγ distance by 0.3 Å[26]. While our conclusions were identical, the effect could be observed much more clearly with GaMD: the L435F mutation increased the average Cγ-Cγ distance from $7.3 \pm 1.9$ Å in WT γ-secretase to $9.1 \pm 1.2$ Å in the L435F PS1 FAD mutant. Furthermore, notable changes in the conformational spaces of PS1-APP interactions were found in all six PS1 FAD mutants, which were consistent with previous experimental and computational results[10,13,14,16–18,26].

The G384A PS1 mutant was the only exception where no stable "Active" low-energy conformational state was sampled even though biochemical experiments showed that this FAD mutant should have similar proteolytic activity to the L286V PS1 FAD mutant (Fig. 1b, h). Given the immediate adjacent location of G384 to the protonated catalytic aspartate D385, its mutation to a slightly larger residue (glycine to alanine) was expected to disrupt the interaction between PS1 residue D385 and APP residue L49 and even increase the D257–D385 distance. The "I2" and "I3" low-energy conformational states were identified in the free energy profile of the G384A mutant (Fig. 6d). The mutant also sampled the "Active" conformation with hydrogen bond formed between PS1 residue D385 and APP residue L49 and ~7–8 Å distance between the PS1 catalytic aspartates, although its probability was not high enough to appear as a low-energy state. The discrepancy here could result from potential inaccuracy of the force field parameters and/or still insufficient sampling of the large enzyme-substrate complex. Moreover, as the pKa value of D257 was reasonably close to that of D385 (7.95 vs. 8.80) (Supplementary Table 2), there could be possible proton exchange between the two catalytic aspartates that could not yet be simulated. Furthermore, we could not determine the Aβ49/Aβ48 ratio quantitatively from the GaMD simulations in this study. While the ratio of AICD50-99 of AICD49-99 was measured at ~1.1 ± 0.1 from MS experiments of the WT APP-bound γ-secretase (Fig. 1), the ratio between Aβ49 and Aβ48 produced from WT APP-bound γ-secretase in natural cell lines is ~7:3[55]. Nevertheless, the experiments were still proceeded as our focus was to determine the relative differences in the quantities of AICD produced between WT and PS1 FAD mutants. We also mainly examined GaMD free energy profiles between the WT and FAD mutants of PS1.

In conclusion, we have presented the dynamic models for cleavage of amyloid precursor protein (APP) by PS1 FAD mutants of γ-secretase, which were consistent with MS and western blotting biochemical experiments. Our findings were also in good agreement with Chen et al. and others[17,18,22,26,43,51,52], even though the effects were clearer due to the enhanced sampling power of GaMD. First, we found that the PS1 FAD mutants confined the active site in PS1 and APP substrate. Second, the PS1

FAD mutants were found to reduce γ-secretase proteolytic activity by hindering APP residue L49 from proper orientation in the active site and/or disrupting the distance between the catalytic aspartates. Our findings here provided mechanistic insights into how PS1 FAD mutants affect structural dynamics and enzyme-substrate interactions of γ-secretase and APP.

## Materials and methods

**C100-FLAG purification**. *E. coli* BL21 cells were grown shaking in LB media at 37 °C until $OD_{600}$ reached 0.6. Cells were induced with 0.5 mM IPTG and were grown for 4 h. The cells were collected by centrifugation and resuspended in 50 mM HEPES pH 8, 1% Triton X-100. The cell suspension was passed through French press to lyse the cells and the lysate was incubated with anti-FLAG M2-agarose beads from SIGMA. Bound substrates were then eluted from the beads with 100 mM Glycine pH 2.5, 0.25% NP-40 detergent and then neutralized with Tris HCl prior to being stored at −80 °C.

**Generation of tetracistronic γ-secretase FAD mutant constructs**. Four monocistonic pMLINK vectors, each encoding one of the γ-secretase components (pMLINK-PS1, pMLINK-Aph1, pMLINK-NCT and pMLINK-Pen-2), were obtained courtesy of Prof. Yigong Shi (Tsinghua University, Beijing, China). Monocistonic pMLINK-PS1 vector was mutated using Platinum™ Superfi II mutagenesis kit (Invitrogen). All constructs were verified by sequencing by ACGT. Each vector had LINK1 and LINK2 sequence flanking the gene of interest. LINK1 contains Pac1 restriction site and LINK2 has PacI and Swa1 restriction site. Mutated monocistronic pMLINK-PS1 vector was treated with restriction enzyme Pac1 to release the gene of interest (PS1). Similarly, pMLINK-APh1 was treated with SwaI restriction enzyme to linearize the vector. The released PS1 from PacI digestion was inserted into linearized pMLINK-Aph1 by ligation independent cloning (LIC) to create bicistronic pMLINK-Aph1-PS1. Similarly, bicistonic pMLINK-Pen2-Nicatrin was created using LIC method. Finally, the two bicistronic vectors were used to make the final tetacistronic vector (pMLINK-PEN-2-nicas-trin-APH-1-PS1) by LIC method.

**γ-secretase expression and purification**. γ-secretase was expressed in HEK 395 F cells by transfection with tetracistronic WT and FAD mutant pMLINK vector. For transfection, HEK 393 F cells were grown in unsupplemented Freestyle 293 media (Life Technologies, 12338-018) until cell density reached $2 \times 10^6$ cells/ml. 150 mg of vector was mixed with 450 mg of 25 kDa linear polyethylemimines (PEI) and incubated for 30 min at room temperature. The DNA-PEI mixtures were added to HEK cells and cells were grown for 60 h. The cells were harvested, and γ-secretase was purified as described previously[56].

**In vitro γ-secretase assay and immunoblotting of AICD products**. 30 nM of WT or FAD mutant γ-secretase was preincubated for 30 min at 37 °C in assay buffer composed of 50 mM HEPES pH 7.0, 150 mM NaCl, and 0.25% 3-[(3-cho-lamidopropyl) dimethylammonio]-2-hydroxy-1-propanesulfonate (CHAPSO), 0.1% phosphatidylcholine and 0.025% phosphatidylethanolamine. Reactions were initiated by addition of purified 3 mM C100-FLAG substrate[57] and incubated at 37 °C for 16 h. The reactions were stopped by flash freezing in liquid nitrogen and stored at −20 °C. Stored γ-secretase reaction mixtures and C100-FLAG standards were subjected to SDS-PAGE on 4-12% bis-tris gels and transferred to PVDF membranes. Membranes were blocked with 5% dry milk for 1 h at ambient temperature and treated with anti-Flag M2 antibodies (SIGMA) for 16 h at 4 °C. Then the blot was washed and incubated with anti-mouse secondary antibodies for 1 h at ambient temperature. The membrane was washed and imaged for chemilumi-nescence, and bands were analyzed by densitometry.

**Detection of AICD species**. AICD-FLAG produced from the enzymatic assay were isolated by immunoprecipitation with anti-FLAG M2 beads (SIGMA) in 10 mM MES pH 6.5, 10 mM NaCl, 0.05% DDM detergent for 16 hours at 4 °C. AICD products were eluted from the anti-FLAG beads with acetonitrile:water (1:1) with 0.1% trifluoroacetic acid. The elutes were run on a Bruker autoflex MALDI-TOF mass spectrometer.

**Simulation system setup**. The cryo-EM structure of APP-bound γ-secretase (PDB: 6IYC)[36] was used to prepare the simulation systems. Two artificial disulfide bonds between C112 of PS1-Q112C and C4 of PS1-V24C were removed as the WT residues (Q112 and V24) were restored. Five unresolved residues at the N-terminus of APP substrate C83 were added through homology modeling by SWISS-MODEL[58]. The large missing hydrophilic loop that connected TM6a and TM7 was not modelled as in our previous studies[30,31], which had no noticeable effects on our final results. In fact, the large missing hydrophilic portion that connects TM6a and TM7 is missing in the cryo-EM structure[36], but this region is not conserved and does not contain sites of PS1 FAD mutations. Moreover, Gopal Thinakaran's lab demonstrated years ago that this region is unnecessary for presenilin proteolytic function[59]. This is in contrast to the hydrophobic region of loop 6, which is

conserved, critical for function, and a domain with many PS1 FAD mutations[59]. The autoproteolytic cleavage of this loop upon assembly of presenilin with the other components of the γ-secretase complex results in the functional protease. The hydrophobic portion of the cleaved loop 6 becomes the TM6a region that is folded into the structure of γ-secretase[36]. The hydrophilic region, now the N-terminus of the presenilin CTF subunit generated by autoproteolysis, is not visible by cryo-EM, even with bound substrate, presumably because it is unstructured and not folded into the active protease complex[36]. The starting structure of WT APP-bound γ-secretase was provided in Supplementary Data 1. Selected PS1 FAD mutations, including P117L, I143T, L166P, G384A, L435F, and L286V (Fig. 1a), were computationally generated using the *Mutation* function of CHARMM-GUI[60–66]. Furthermore, residue D385 in PS1 was protonated to simulate γ-secretase activation for ε cleavage of APP based on the results of PROPKA3 calculations[53,54] (Supplementary Table 1). All chain termini were capped with neutral patches (acetyl and methylamide). The enzyme-substrate complexes were embedded in POPC membrane lipid bilayers and then solvated in 0.15 M NaCl solutions using the CHARMM-GUI webserver[60,62–67].

**Simulation protocols**. The CHARMM36m force field parameter set[68] was used for the protein and lipids. The simulation systems were initially energetically minimized for 5000 steps using the steepest-descent algorithm and equilibrated with the constant number, volume, and temperature (NVT) ensemble at 310 K. They were further equilibrated for 375 ps at 310 K with the constant number, pressure, and temperature (NPT) ensemble. Short cMD simulations were then performed for 10 ns using the NPT ensemble with constant surface tension at 1 atm pressure and 310 K temperature. GaMD implemented in the GPU version of AMBER 20[27,69] was applied to simulate the effects of PS1 FAD mutations on γ-secretase activation for ε cleavage of APP. The simulations involved an initial short cMD of 15 ns to calculate GaMD acceleration parameters and GaMD equilibration of added boost potentials for 60 ns. Three 1000–1500 ns independent all-atom dual-boost GaMD production simulations with randomized initial atomic velocities were performed on the APP-bound γ-secretase complexes, with the reference energy set to lower bound. The upper limits of the boost potential standard deviations, $\sigma_{0P}$ and $\sigma_{0D}$, were set to 6.0 kcal/mol for both dihedral and total potential energetic terms. The GaMD simulations are summarized in Supplementary Table 2.

**Simulation analysis**. The simulation trajectories were analyzed using VMD[70] and CPPTRAJ[38]. The distance between Cγ atoms of catalytic aspartates PS1-D257 and D385 and distance between PS1 residue D385 (atom OD2) and APP residue L49 (atom O) were calculated. The PyReweighting[28] toolkit was applied for free energy calculations from the D257 – D385 and D385 – L49 distances for each system (Fig. 1). A bin size of 1 Å and cutoff of 500 frames in each bin was used to calculate the two-dimension (2D) potential mean force (PMF) free energy profiles. The time courses of APP secondary structures were calculated by CPPTRAJ[38]. Simulation frames were saved every 1 ps. The hierarchical agglomerative structural clustering algorithm in CPPTRAJ[38] was performed on GaMD simulations of WT, P117L, I143T, L166P, G384A, L435F, and L286V PS1 FAD mutant APP-bound γ-secretase to identify representative poses for low-energy conformational states (Supplementary Data 1).

**Reporting summary**. Further information on research design is available in the Nature Portfolio Reporting Summary linked to this article.

# Data availability
Data supporting the findings of this study are included in the article and its Supplementary Information and Data files.

# Code availability
This study utilized the standard builds of the simulation software AMBER 20 (https://ambermd.org) according to best practices for running GaMD simulations[27] with all parameters specified in the Methods section.

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

## Acknowledgements

This work used supercomputing resources with allocation award TG-MCB180049 through the Extreme Science and Engineering Discovery Environment (XSEDE), which is supported by National Science Foundation grant number ACI-1548562, and project M2874 through the National Energy Research Scientific Computing Center (NERSC), which is a U.S. Department of Energy Office of Science User Facility operated under Contract No. DE-AC02-05CH11231, and the Research Computing Cluster at the University of Kansas. This work was supported in part by the startup funding in the College of Liberal Arts and Sciences at the University of Kansas and award 2121063 from National Science Foundation (Y.M.) and AG66986 from the National Institutes of Health (M.S.W.).

## Author contributions

H.N.D. performed GaMD simulations, analyzed simulation data and wrote the manuscript. S.D. performed mutagenesis, western blotting and mass spectrometry experiments and wrote the manuscript. A.B. interpreted data. Y.M. and M.S.W. supervised the project, interpreted data and wrote the manuscript. All authors contributed towards the final version of the manuscript.

## Competing interests

The authors declare no competing interests.
