## [Peer Review File · Communications Biology]

Reviewers' comments:

Reviewer #1 (Remarks to the Author):

Miao, Wolfe and co-workers present in this manuscript an investigation on how the FAD mutations on PS1 impact the gamma-secretase activation for cleaving APP through the enhanced sampling GaMD simulations and biochemical experiments. The study is well carried out and the results are presented in a clear and concise way. Given the importance of this topic, it should be appropriate for publication in the Communications Biology. The manuscript could be further improved by addressing the following comments.

1) As shown in Figure 1 D-J, most mutation systems have only one minimum, and even for those have multiple minima/states, the free energy barriers between minima are rather small (less than 3 kcal/mol). Such a barrier should be able to cross even with a few hundred nanoseconds of conventional MD simulations. Given that the GaMD are carried out with 3 replicas, each 1000 - 1500 ns, I would highly suggest the authors to carry out some 1000 ns conventional MD simulations to compare the free energy landscapes. With that the authors should be able to demonstrate how GaMD enhanced the sampling in this particular problem, or alternatively, exclude the case that even with GaMD the simulation systems were not accessing relevant conformational states.

2) The exact definition of D385 - L49 distance (which atom of L49) should be provided in the main text, as this was used, together with the D287-D385 distance, to infer the mechanism of the cleavage reaction. The rationale to use these two CVs should also be discussed in more details.

3) The free energy profiles for D385-L49 and D257-D385 were constructed, which correspond to the major configuration to generate AICD 50-99. How about those for D385-T48 and D257-D385, which corresponds to AICD 49-99?

4) The experimental ratios between AICD50-99 and AICD49-99 were calculated from the signal intensities corresponding to the two products. How were the uncertainties determined? This could be important as for the wildtype the result differed significantly from the ratio reported in previously published work (J. Am. Chem. Soc. 2022, 144, 14, 6215-6226).

5) The author proposed whether the water involved coordination with the catalytic aspartates can be inferred based on the distance between the C γ atoms of D257 and D385 (lines 216, 280, 284, etc). As explicit solvent MD simulations were carried out, it might be better to also provide more direct structural information by analyzing the water dynamics between the catalytic aspartates.

6) In the analysis of secondary structures of APP substrates, the authors made statements like "the N-terminus of APP (residues L17-K28) became more structured, with most of the time spent as turns". I think for APP, there are only helical and unstructured states. I don't think turns or beta-bridges are really meaningful secondary structures in this particular case.

7) Line 235, the definition of "P1', P2', and P3' residues and corresponding S1', S2', and S3' subpockets" is not clear.

Reviewer #2 (Remarks to the Author):

The paper reports a detailed and mostly well-performed computational + MALDI-TOF study of the conformational ensembles of PS1 mutants of gamma secretase and cleavage products, a topic of substantial interest e.g., to treatments of Alzheimer's disease based on gamma secretase modulators. I think the paper deserves publication in a technical journal, but I do not see any novelty of the study, which also has some methodological and presentation issues that need to be addressed, and would recommend either major revision or transfer to a more suitable journal. Regardless the paper and structures produced need to be reviewed after major revision.

Specifically:

Regarding novelty and presentation:

1) In general, there is a lot of selective citation in this paper, although the authors are clearly inspired by previous work not cited, which always leaves a bad impression. A few examples:

1a. They refer to the "proteasome of the membrane" without citing Kopan et al.
<https://www.nature.com/articles/nrm1406>

1b. The most detailed and complete experimental analysis of PS1 cleavage products by Sun et al. in PNAS (cited +200 times) is not cited – I wonder why, this work showed that enzyme activity and 42/40 was a very general thing not confined e.g., to the 5-10 most commonly studied mutations. The authors need to discuss this work
<https://www.pnas.org/doi/abs/10.1073/pnas.1618657114>

1c. The mechanism described seems to have been already been identified in a large body of previous experimental and theoretical work dating back to 2015 that is not cited in the paper but reviewed in WIREs Comput. Mol. Sciences
<https://wires.onlinelibrary.wiley.com/doi/abs/10.1002/wcms.1556>

1d. There is a notorious shortage on important recent experimental work from Harald Steiner and co-workers on this problem, while citing a lot of not very important simulation papers by the authors themselves.

1e. I suggest the authors read and cite this review by Hitzzenberger et al. and some of the work therein to provide a more balanced assessment of the topic and field.
<https://www.sciencedirect.com/science/article/pii/S108495211830274X>

2) MD simulations can give many different things due to errors in system realism, force field, sampling etc. Thus comparison to experimental structures is absolutely essential. The authors need to show that they can regenerate the experimental structure s Asp-Asp distance which is long under conditions of their simulations, as the very first thing, and then move on from there. A lot of the Asp-Asp- distances are very short and not seen ever experimentally to my knowledge.

3) The authors mention age of onset of the mutants but do not say where they got the numbers – they need reference. Also, what is the clinical relevance of the 42/40 ratio? If it relates to disease one could expect a relationship to clinical outcome, since the wild type is apparently sporadic AD with age of onset perhaps 75 years or later, the PS1 mutants are down to 40 years or so – the study by Tang and Kepp (Journal of Alzheimer's Disease 66 (2018) 939–945) shows that the data by Sun et al. actually correlate with age of onset, making the 42/40 ratio of clinical relevance (not necessarily causal but at least correlative). I suggest discussing this, as they age of onset right now pops out of the blue.

4) The authors need to present their data in the context of previous simulations and explain why they think theirs are better or at least complementary. Just picking two previous simulation studies, one by Zacharias' group and one by the Kepp group and use one-liners to dismiss them without acknowledging weakness in their own approach (see below) is not enough. There are weaknesses in all approaches and the novelty and variations and new insight gained here needs to be more clear.

5) What is the evidence that one aspartate needs to be protonated? This is not explained / cited. The use of one protonated aspartate reduces Asp-Asp repulsion and is probably partly responsible for the very short distances seen in this paper, where the two aspartates almost come in contact – this is unprecedented to my knowledge, and no experimental or simulated data suggest so – it could easily be an artefact of having protonation on one aspartate. Remember that aspartate needs to be deprotonated to take a proton from the nucleophilic water to enhance its nucleophilicity. This whole topic needs more discussion. A suspicion is that the states here are too

compact because the experimental cryo-EM structures are more like 11 Å Asp-Asp distance (should be compared in earnest).

6) Table S2. The authors need to explain why the simulations vary in length, with some mutants simulated for longer and some for less – if they are not treated equally how can you be sure your comparison is not confounded by simulation extent?

7) On a similar note (and related to point 2), a direct comparison of WT vs. mutants in context of the intra-system variation (error bars/Precision) seen from the three triplicates is expected – this is normal standard to identify if your conformation tendencies are significantly different in the mutants and WT. For example you could use bar plots with SD to quantify the distances between Aspartates or to the substrate in WT vs. mutants. Right now, comparisons appear sporadic, which is always prone to selection bias. See Figure S2 and S3: The noise in the graphs make any difference between the systems less significant, and multiple simulations would help to determine if this is the case.

8) The initial structures and the final representative cluster structures (from equilibrated parts of the trajectories) should be uploaded as PDB files, otherwise it is impossible to understand and reproduce this work and determine the quality and realism of the models applied.

9) Many of the central aspects used in the analysis are drawn from already published work, which has evidently been read but not cited. Such as the monitoring of Asp-Asp and Asp-substrate distances, without reference and comparison to earlier work, the use of names such as “semi-closed” conformation is confusing since other groups used the same name or “semi-open” for perhaps other conformation states. All these similar names are confusing to the outsider if not clearly defined by some metric, like e.g. the Asp-Asp-distance or the size of the catalytic pocket, as attempted by the Kepp group.

10) A lot of the results section is way too long and insignificant given the uncertainties in finer details of MD structures. The authors speculate on energies and structures with two decimal precision when the real uncertainty is probably in the full kcal/mol unit scale. The authors really need to get a sense of proportion not to tire the reader with details that are not important. The central part is general agreement with experiment in relevant limits / conditions, show effect of mutations, compare to previous simulations, define clearly and quantitatively the few much chemically important states, forget about small Å displacements and two-decimal energy effects, and then explain better the novelty and significance.

11) Gamma secretase is a very slow enzyme. (kcat for APP ε proteolysis ~ 2-6 per hour) Do the effects here translate into a KM or kcat effect in the enzyme? This really needs to be discussed in contrast to many of the unimportant details that take space now.

Regarding methodology:

1) The authors should show for some mutations studied before by other labs that their cleavage processing effects are consistent with those of other groups and not lab/protocol dependent, as one can imagine lab specific variations. Always essential to establish agreement and consensus, both to previous simulations and previous experiments.

2) The rationale for using a protonated Asp needs to be explained as it could explain why their ensembles produce much shorter Asp-Asp distances (down to few Å, suggesting almost direct contact between a protonated and non-protonated Asp) than seen experimentally (10-12 Å). In that light, dismissing previous work due to the use of two deprotonated aspartates without explanation of references to the need for one to be protonated is also not enough. The claim that the enzyme cannot be active with two deprotonated Asp seems incorrect (page 4).

3) Also previous work studying the gamma secretase cryo-EM structures under various conditions also studied PH effects with PropKa and found results distinct from those here – “only at pH 5 did a protonated state appear for the enzyme complex.”

<https://pubs.rsc.org/en/content/articlelanding/2020/cp/c9cp06723j/unauth>

This needs to be discussed and explained, as the deprotonation may explain a lot of the unusual compact structures in the present work.

4) The authors forget to tell what they did with the large hydrophilic loop, which is not in the cryo-EM structure they start out from, but which is present in the real system and thus experimental / biological relevant data. So are you comparing apples and perries? The large loop combines TM6 and 7 and could have major influence on modulating the active site, as previous work suggests. The absence of the loop could also reduce hydrophobic repulsion between the NTF and CTF loop parts after maturation which then could also, like the deprotonation, contribute to the very short Asp-Asp distances observed.

5) The authors use an MD method that introduces approximations to the sampling by a Gaussian function, but the impact of this approximation is not shown by comparison to a full conventional simulation for this protein. It is relevant to show for gamma secretase specifically because of its very loose conformational states and many (potentially non-Gaussian) modes. This again illustrates that the present simulations are different from, but not necessarily better than previous simulations, yet this comparison and discussion is missing in the paper.

6) Previous work found that the substrate helix partly unzips when in the pocket – any evidence for this would be relevant to discuss in relation to how the nucleophilic water accesses the peptide bonds consecutively.

7) The seven lines of conclusion on page 22-23 are very vague and it is not clear what is old and what is new, compared to previous insights. After going through all the above aI believe it will be easier for the authors to write a conclusion that provides strong information on what actually is gained at the molecular level from this paper.

In summary, the paper's work is substantial but also not very novel and the presentation is very incomplete, and several points of the methodology need substantial further attention. I recommend major revision and probably submission to a more technical journal although Commun. Biol is also possible if the advice above is followed and a lot of less relevant discussion on pages 8-22 is replaced with more direct assertive comparison to experimental cryo-EM structures and previous simulation work, showing the key molecular insights in better focus.

Minor points:

1) it is very hard to see the differences claimed in Figure S5

2) the use of names for conformations is complicated and a lot of this discussion could be simplified.

Reviewer #3 (Remarks to the Author):

It is very important to understand how FAD mutants of PS1 affect structural dynamics of gamma-secretase complex and substrate, in terms of Abeta42 production. In this study, authors performed structural simulation study and biochemical analyses on several FAD mutants of PS1 and APP derived substrate. Although one may consider that simulation study is just a prediction, it helps us to gain insights into how this enzyme produces Abeta42. This reviewer has several comments on this manuscript.

1. Please include the following references in the introduction part of tricarboxypeptidase trimming, because this is the 1st paper to demonstrate it: gamma-Secretase: successive tripeptide and tetrapeptide release from the transmembrane domain of beta-carboxyl terminal fragment. Takami M, Nagashima Y, Sano Y, Ishihara S, Morishima-Kawashima M, Funamoto S, Ihara Y. *J Neurosci*. 2009 Oct 14;29(41):13042-52. doi: 10.1523/JNEUROSCI.2362-09.2009.
2. Quantification of AICD No.1: It seems that AICD bands on I143T, L166P, and L435F are not detected at all (Fig. 1B). If so, these are out of range of the standard curve? This reviewer wonders how the authors quantified AICD levels
3. Quantification of AICD No.2: The authors used C100, as an authentic standard to quantify AICD concentration. However, this reviewer wonders whether retention level of AICD on PVDF membrane is same as that of C100, or not. Because AICD is much smaller than C100, so retention level of AICD on the membrane is less. The best way to quantify AICD on the membrane is to use AICD as authentic standard, not C100.
4. This reviewer wonders whether 2D free energy profiles with distance between PS1 Asp385 and C100 Val50 are similar to Fig. 1E-J. Any reason why no simulation with Asp385-Val50?
5. This reviewer is curious to know when C100 gets cleaved in time course studies in Fig. 5.

Reviewers' comments:

We thank the reviewers for the supportive and constructive comments. The reviewers' comments are addressed pointwise below, with changes to the text highlighted in yellow.

Reviewer #1 (Remarks to the Author):

Miao, Wolfe and co-workers present in this manuscript an investigation on how the FAD mutations on PS1 impact the gamma-secretase activation for cleaving APP through the enhanced sampling GaMD simulations and biochemical experiments. The study is well carried out and the results are presented in a clear and concise way. Given the importance of this topic, it should be appropriate for publication in the Communications Biology. The manuscript could be further improved by addressing the following comments.

1) As shown in Figure 1D-J, most mutation systems have only one minimum, and even for those have multiple minima/states, the free energy barriers between minima are rather small (less than 3 kcal/mol). Such a barrier should be able to cross even with a few hundred nanoseconds of conventional MD simulations. Given that the GaMD are carried out with 3 replicas, each 1000 - 1500 ns, I would highly suggest the authors to carry out some 1000 ns conventional MD simulations to compare the free energy landscapes. With that the authors should be able to demonstrate how GaMD enhanced the sampling in this particular problem, or alternatively, exclude the case that even with GaMD the simulation systems were not accessing relevant conformational states.

Response: As suggested by the Reviewer, we have performed additional conventional MD (cMD) simulations (1500 ns x 4) on each of four representative APP-bound γ -secretase systems, including the WT, P117L, I143T, and L166P PS1 FAD mutants. The new cMD simulation results compared with GaMD simulations have been described in the 2nd paragraph on page 20 of the Discussion section in the revised manuscript as "...We performed four additional 1.5 μ s cMD simulations on each of four representative APP-bound γ -secretase systems, including the WT and the P117L, I143T, and L166P PS1 FAD mutants. The time courses of the D257-D385, D385-V50, D385-L49, and D385-T48 distances calculated from the cMD simulations were plotted in **Figure S15**. 2D free energy profiles of the (D257-D385, D385-L49), (D257-D385, D385-V50), or (D257-D385, D385-T48) distances (**Figure S16**) were calculated and compared with those from GaMD simulations (**Figure 1** and **S17**). For both the cMD and GaMD simulations, the low-energy conformational states calculated from both the D385-V50 and D385-T48 distances matched those calculated from the D385-L49 distances. Moreover, GaMD sampled larger conformational space than the cMD simulations and uncovered additional low-energy conformational states in the WT, I143T, and L166P FAD mutant γ -secretase systems (**Figures 1** and **S17**). In particular, the WT, I143T, and L166P simulation systems visited two ("Inhibited" and "I1"), one ("I4"), and one ("I3") additional low-energy conformational states in the GaMD simulations than in the cMD simulations, respectively (**Figures 1D,F,G** and **S16A,C,D**). In the P117L simulation system, both GaMD and cMD uncovered two low-energy conformational states, i.e., the "Active" and "I2" (**Figures 1E** and **S16B**). These findings demonstrated the enhanced sampling power of GaMD in simulations of large biomolecules such as γ -secretase...."

2) The exact definition of D385 - L49 distance (which atom of L49) should be provided in the main text, as this was used, together with the D287-D385 distance, to infer the mechanism of the cleavage reaction. The rationale to use these two CVs should also be discussed in more details.

Response: The exact atoms from the 6IYC PDB structures were included in the 1st paragraph on page 9 of the Results section in the revised manuscript as "...Therefore, the distance between the C γ atoms of catalytic aspartates D257 and D385 in PS1 and the distance between PS1 residue D385 (protonated oxygen) and APP residue L49 (carbonyl oxygen) were calculated from the GaMD simulations and plotted in **Figures S2–S5**....". The rationale to use these two CVs was explained on the same page as "...The ϵ -cleavage of APP by γ -secretase can only be carried out when the two PS1 catalytic aspartates are at a suitable distance so that a nucleophilic water molecule can be recruited for the proteolytic reaction through water-bridged hydrogen bonding with the two aspartates^{5,26,30}. Furthermore, the carbonyl group at the cleavage site on APP (residue L49) would form another hydrogen bond between the carbonyl oxygen and protonated carboxylic side chain of catalytic residue D385 in PS1 for proteolysis^{5,26,30}...."

3) The free energy profiles for D385-L49 and D257-D385 were constructed, which correspond to the major configuration to generate AICD 50-99. How about those for D385-T48 and D257-D385, which corresponds to AICD 49-99?

Response: As suggested by the Reviewer, we have calculated the 2D free energy profiles of the D257-D385 and D385-T48 distances as shown in the new **Figure S17**. Since the main focus of this study was to explore the differences in proteolytic activity between PS1 FAD mutant and WT γ -secretase with APP, these free energy profiles were described briefly starting from the 2nd paragraph on page 20 as "...2D free energy profiles of the (D257-D385, D385-L49), (D257-D385, D385-V50), or (D257-D385, D385-T48) distances (**Figure S16**) were calculated and compared with those from GaMD simulations (**Figure 1** and **S17**). For both the cMD and GaMD simulations, the low-energy conformational states calculated from both the D385-V50 and D385-T48 distances matched those calculated from the D385-L49 distances. Moreover, GaMD sampled larger conformational space than the cMD simulations and uncovered additional low-energy conformational states in the WT, I143T, and L166P FAD mutant γ -secretase systems (**Figures 1** and **S17**). In particular, the WT, I143T, and L166P simulation systems visited two ("Inhibited" and "I1"), one ("I4"), and one ("I3") additional low-energy conformational states in the GaMD simulations than in the cMD simulations, respectively (**Figures 1D,F,G** and **S16A,C,D**). In the P117L simulation system, both GaMD and cMD uncovered two low-energy conformational states, i.e., the "Active" and "I2" (**Figures 1E** and **S16B**). These findings demonstrated the enhanced sampling power of GaMD in simulations of large biomolecules such as γ -secretase...."

In this study, we could not determine the A β 49/A β 48 ratio quantitatively from the GaMD simulations for comparison with the experimental data. We explained the reason in the 1st paragraph on page 26 as "...Furthermore, we could not determine the A β 49/A β 48 ratio quantitatively from the GaMD simulations in this study. While the ratio of AICD50-99 of AICD49-99 was measured at $\sim 1.1 \pm 0.1$ from MS experiments of the WT APP-bound γ -secretase (**Figure 1**), the ratio between A β 49 and A β 48 produced from WT APP-bound γ -secretase in natural cell lines is $\sim 7:3$ ⁵⁵. Nevertheless, the experiments were still proceeded as our focus was to determine the relative differences in the quantities of AICD produced between WT and PS1 FAD

mutants. We also mainly examined GaMD free energy profiles between the WT and FAD mutants of PS1....”

4) The experimental ratios between AICD50-99 and AICD49-99 were calculated from the signal intensities corresponding to the two products. How were the uncertainties determined? This could be important as for the wildtype the result differed significantly from the ratio reported in previously published work (J. Am. Chem. Soc. 2022, 144, 14, 6215–6226).

Response: The MALDI-TOF experiments to determine signal intensities corresponding to AICD50-99 and AICD49-99 were done in triplicates. The average and standard deviation of the 3 ratios were reported for each γ -secretase system in **Figure 1C**. For more clarity, we performed the t-tests and added the p values along with the ratios in **Figure 1C** to highlight the significance of the ratios determined for the PS1 FAD mutants. The ratio for the WT determined in our J. Am. Chem. Soc. 2022, 144, 14, 6215-6226 fell within the average \pm standard deviation of the ratio determined in this current manuscript. Additional clarifications were added to Figure 1 caption on page 37 as “...T-tests were performed, and the resulting p values were added along with the ratios to highlight the significance of the ratios determined for the PS1 FAD mutants....”

5) The author proposed whether the water involved coordination with the catalytic aspartates can be inferred based on the distance between the C γ atoms of D257 and D385 (lines 216, 280, 284, etc). As explicit solvent MD simulations were carried out, it might be better to also provide more direct structural information by analyzing the water dynamics between the catalytic aspartates.

Response: Because exceedingly large simulation files were generated for coordinates of all atoms in the γ -secretase systems, we saved atomic coordinates for only the protein and substrate in the Amber simulation trajectories. Nevertheless, as suggested by the Reviewer, we reproduced a 100ns GaMD simulation of the “Active” WT γ -secretase, starting from the 1200ns checkpoint of Sim1, and saved the coordinates of not only the proteins and substrates but also lipids, ions, and water molecules. The water dynamics during this simulation was examined in **Figure S10**. Our detailed observations were included in the 2nd paragraph on page 14 as “...To further examine the water dynamics during γ -secretase activation for ϵ -cleavage of APP, we reproduced a 100ns GaMD simulation of the “Active” WT γ -secretase, starting from the 1200ns checkpoint of Sim1, and saved the coordinates of not only proteins and substrates but also lipids, ions, and water molecules (**Figure S10**). The time courses of the D257-D385 and D385-L49 distances were calculated and shown in **Figure S10A**. Upon the formation of the D385-L49 hydrogen bond at $\sim 3\text{\AA}$ distance while the PS1 residues D257 and D385 maintained $\sim 6\text{-}8\text{\AA}$ distance, a water molecule was recruited (**Figure S10B**) and trapped between the two catalytic aspartates (**Figure S10C**) to carry out the proteolytic reaction in the “Active” conformation. This has also been observed in our previous study³⁰....”

6) In the analysis of secondary structures of APP substrates, the authors made statements like "the N-terminus of APP (residues L17–K28) became more structured, with most of the time spent as turns". I think for APP, there are only helical and unstructured states. I don't think turns or beta-bridges are really meaningful secondary structures in this particular case.

Response: We agree with the reviewer that turns and β -bridges might not be meaningful secondary structures for APP in the N-terminus. We have removed the related descriptions in the revised manuscript.

7) Line 235, the definition of “P1’, P2’, and P3’ residues and corresponding S1’, S2’, and S3’ subpockets” is not clear.

Response: We included the definitions of P1’, P2’, P3’ residues and corresponding S1’, S2’, and S3’ subpockets in the 3rd paragraph on page 12 as “...Here, P1’, P2’, and P3’ referred to APP residues that were one, two, and three residues away downwards, respectively, from the APP cleavage side residue L49 (i.e., V50, M51, and L52). The corresponding S1’, S2’, and S3’ subpockets consisted of residues that were within 5 Å of APP substrate residues P1’ V50, P2’ M51, and P3’ L52....”

Reviewer #2 (Remarks to the Author):

The paper reports a detailed and mostly well-performed computational + MALDI-TOF study of the conformational ensembles of PS1 mutants of gamma secretase and cleavage products, a topic of substantial interest e.g., to treatments of Alzheimer’s disease based on gamma secretase modulators. I think the paper deserves publication in a technical journal, but I do not see any novelty of the study, which also has some methodological and presentation issues that need to be addressed and would recommend either major revision or transfer to a more suitable journal. Regardless the paper and structures produced need to be reviewed after major revision. Specifically:

Regarding novelty and presentation:

1) In general, there is a lot of selective citation in this paper, although the authors are clearly inspired by previous work not cited, which always leaves a bad impression. A few examples:

1a. They refer to the “proteasome of the membrane” without citing Kopan et al. <https://www.nature.com/articles/nrm1406>

Response: We have moved the term “proteasome of the membrane” from the Abstract to the Introduction and cited the reference in the 1st paragraph on page 3 of the Introduction section as “...PS1 is the catalytic component of γ -secretase, “the proteasome of the membrane”³ which carries out intramembrane proteolysis of more than 150 peptide substrates⁴, including amyloid precursor protein (APP), via two conserved aspartates, D257 and D385^{5,6}...”

1b. The most detailed and complete experimental analysis of PS1 cleavage products by Sun et al. in PNAS (cited +200 times) is not cited – I wonder why, this work showed that enzyme activity and 42/40 was a very general thing not confined e.g. to the 5-10 most commonly studied mutations. The authors need to discuss this work <https://www.pnas.org/doi/abs/10.1073/pnas.1618657114>

Response: We have included a summary of the work starting from the end of the 2nd paragraph on page 3 of the Introduction as “...Recently, Sun et al. analyzed 138 pathogenic mutations in the

PS1 of γ -secretase on the in vitro production of A β 42 and A β 40 peptides¹⁷. They found that ~90% of the mutations reduced the production of A β 42 and A β 40, and ~10% of these mutations decreased the A β 42/A β 40 ratio¹⁷....”

1c. The mechanism described seems to have already been identified in a large body of previous experimental and theoretical work dating back to 2015 that is not cited in the paper but reviewed in WIREs Comput. Mol. Sciences

<https://wires.onlinelibrary.wiley.com/doi/abs/10.1002/wcms.1556>

Response: In most previous studies discussed in the review, the main effect of PS1 FAD mutations in γ -secretase was suggested to destabilize the enzyme-substrate complex, which led to releases of longer A β peptide products. In contrast, the effects we identified in this study was distinctly different. As summarized in the Abstract, “...The P117L and L286V FAD mutants could still sample the “Active” state for substrate cleavage, but with significantly reduced conformational space compared with the wildtype. The other mutants hardly visited the “Active” state. The PS1 FAD mutants were found to reduce γ -secretase proteolytic activity by hindering APP residue L49 from proper orientation in the active site and/or disrupting the distance between the catalytic aspartates.”

We again clarified our main findings in the 1st paragraph on page 19 as “...Here, our main conclusion was that the PS1 FAD mutant γ -secretase stabilized the active sites of the enzyme-substrate complexes, which was distinctly different from previous studies, where the effects of PS1 FAD mutants were found to be destabilizing the enzyme-substrate complexes, causing the earlier releases of longer A β peptides^{10,22,42-45}....” In addition, we have included the relevant references in the 2nd paragraph of page 4 as “...Kong et al.²⁰ performed the first atomistic simulation of isolated PS1 unit in 2015 and found that transmembrane domains (TM) 2, 6, and 9 were highly mobile^{21,22}....” and in the end of the 2nd paragraph on page 21 as “...This finding was highly consistent with previous simulation studies, in which the repeated formations of β -strands in several solvent-exposed regions of presenilin were observed^{22,43,51,52}....”

1d. There is a notorious shortage on important recent experimental work from Harald Steiner and co-workers on this problem, while citing a lot of not very important simulation papers by the authors themselves.

Response: We have removed the following sentences in the Introduction about related GaMD method developments: “Furthermore, GaMD has been combined with replica exchange MD (REMD)^{28,29} to further improve conformational sampling and free energy calculations²⁷. In addition, “selective GaMD” algorithms, including Ligand GaMD (LiGaMD)³⁰, Peptide GaMD (Pep-GaMD)³¹, and Protein-Protein Interaction-GaMD (PPI-GaMD)³² have been developed to enable repetitive binding and dissociation of small-molecule ligands, highly flexible peptides, and proteins within microsecond simulations, which allow for highly efficient and accurate calculations of ligand/peptide/protein binding free energy and kinetic rate constants²⁷. Recently, GaMD has been combined with deep learning and free energy profiling into GLOW to predict molecular determinants and map free energy landscapes of biomolecules³³.”

Furthermore, we have added the relevant references in the 1st paragraph on page 4 as “...Moreover, Trambauer et al. studied seven A β 43-producing PS1 FAD mutants, including M292D, L166P, V261F, Y256S, R278I, G382A, and L435F, and found that A β 43 was produced

in very high levels when the PS1 function was severely impaired¹⁸. Furthermore, alteration of enzyme-C99-substrate interactions were observed in all these mutants, regardless of their effects¹⁸....”, the end of the 1st paragraph on page 25 as “...Furthermore, notable changes in the conformational spaces of PS1-APP interactions were found in all six PS1 FAD mutants, which were consistent with previous experimental and computational results^{10,13,14,16-18,26}....”, and 2nd paragraph on page 26 as “...Our findings were also in good agreement with Chen et al. and others^{17,18,22,26,43,51,52}, even though the effects were clearer due to the enhanced sampling power of GaMD....”

1e. I suggest the authors read and cite this review by Hitzenberger et al. and some of the work therein to provide a more balanced assessment of the topic and field.

<https://www.sciencedirect.com/science/article/pii/S108495211830274X>

Response: We have included the relevant details from the review in the Introduction and cited them accordingly in the 2nd paragraph of page 4 as “...Kong et al.²⁰ performed the first atomistic simulation of isolated PS1 unit in 2015 and found that transmembrane domains (TM) 2, 6, and 9 were highly mobile^{21,22}. In addition, only inactive distances between catalytic aspartates were sampled in the study because of the electrostatic repulsion caused by the negative charges of the two aspartates forming the active site^{20,21}. The coarse-grained simulations of PS1 as part of the γ -secretase complex illustrated that PS1 was much more likely to be activated when either of the catalytic aspartates was protonated²³. This finding was in good agreement with the proposed mechanism of aspartic proteases, which requires one of the catalytic aspartates to act as an acid²⁴. Hitzenberger and Zacharias observed that the active state of PS1 remained stable even in the absence of a substrate as the direct hydrogen bond between protonated D257, D385, and a water bridge was sufficient to stabilize the active form^{21,25}. Furthermore, the transition towards the active state of PS1 was found to involve TM1, TM6, TM7, TM8, and TM9^{21,25}....”

2) MD simulations can give many different things due to errors in system realism, force field, sampling etc. Thus, comparison to experimental structures is absolutely essential. The authors need to show that they can regenerate the experimental structure's Asp-Asp distance which is long under conditions of their simulations, as the very first thing, and then move on from there. A lot of the Asp-Asp-distances are very short and not seen ever experimentally to my knowledge.

Response: We have made a new **Figure S1**, which included the average D257-D385 distances calculated from GaMD simulations of γ -secretase systems, and **Table S3**, which listed the D257-D385/A385 determined from the available PDB structures of γ -secretase. In contrast to the reviewer's assumption, the lowest D257-D385 distance could reach ~ 3.9 Å in the 5FN2 PDB structure, while most of the experimental D257-D385 distances were between ~ 5 Å and ~ 9 Å. The highest D257-D385 distance was ~ 11.5 Å, observed in the 5FN4 PDB structure. Therefore, the D257-D385 distances observed from our GaMD simulations resembled the experimental structures very well.

These observations were clarified starting from the 2nd paragraph on page 8 as “...In addition, the protonation of one catalytic aspartate (D385) allowed us to obtain comparable D257-D385 distances in our GaMD simulations with the available PDB structures of γ -secretase (**Figure S1** and **Table S3**). In particular, the average D257-D385 distances obtained from GaMD simulations were 7.3 ± 1.9 Å for WT, 7.6 ± 1.1 Å for P117L, 8.2 ± 1.6 Å for I143T, 8.7 ± 1.0 Å

for L166P, $8.1 \pm 1.2 \text{ \AA}$ for G384A, $9.1 \pm 1.2 \text{ \AA}$ for L435F, and $7.4 \pm 1.0 \text{ \AA}$ for L286V PS1 FAD mutant γ -secretase (**Figure S1**). Meanwhile, the lowest D257-D385 distance could get to $\sim 3.9 \text{ \AA}$ in the 5FN2³³ PDB structure, while most of the experimental D257-D385 and D257-A385 (in the 6IDF and 6IYC PDB) distances were between $\sim 5 \text{ \AA}$ and $\sim 9 \text{ \AA}$ ³³⁻³⁷. The highest D257-D385 distance was $\sim 11.5 \text{ \AA}$, observed in the 5FN4³³ PDB structure (**Table S3**)....”

3) The authors mention age of onset of the mutants but do not say where they got the numbers – they need reference. Also, what is the clinical relevance of the 42/40 ratio? If it relates to disease one could expect a relationship to clinical outcome, since the wild type is apparently sporadic AD with age of onset perhaps 75 years or later, the PS1 mutants are down to 40 years or so – the study by Tang and Kepp (Journal of Alzheimer’s Disease 66 (2018) 939–945) shows that the data by Sun et al. actually correlate with age of onset, making the 42/40 ratio of clinical relevance (not necessarily causal but at least correlative). I suggest discussing this, as they age of onset right now pops out of the blue.

Response: Since our focus here does not include addressing why different PS1 FAD mutants lead to different ages of onsets, we have removed the ages of onset from the following sentence in the 2nd paragraph on page 6: “We selected six PS1 FAD mutations to investigate based on early age of disease onset and their representative locations relative to the transmembrane domains (TM) of PS1, including P117L (hydrophobic loop 1), I143T (TM2), L166P (TM3), L286V (TM6, active site), G384A (TM7, active site), and L435F (TM9) (**Figure 1A**)....”

4) The authors need to present their data in the context of previous simulations and explain why they think theirs are better or at least complementary. Just picking two previous simulation studies, one by Zacharias’ group and one by the Kepp group and use one-liners to dismiss them without acknowledging weakness in their own approach (see below) is not enough. There are weaknesses in all approaches and the novelty and variations, and new insight gained here needs to be clearer.

Response: We appreciate the reviewer’s comment, but we did point out the weaknesses of our work starting from the 2nd paragraph on page 25 in the original manuscript as “...The G384A PS1 mutant was the only exception where no stable “Active” low-energy conformational state was sampled even though biochemical experiments showed that this FAD mutant should have similar proteolytic activity to the L286V PS1 FAD mutant (**Figure 1B** and **1H**). Given the immediate adjacent location of G384 to the protonated catalytic aspartate D385, its mutation to a slightly larger residue (glycine to alanine) was expected to disrupt the interaction between PS1 residue D385 and APP residue L49 and even increase the D257–D385 distance. The “I2” and “I3” low-energy conformational states were identified in the free energy profile of the G384A mutant (**Figure 6D**). The mutant also sampled the “Active” conformation with hydrogen bond formed between PS1 residue D385 and APP residue L49 and $\sim 7\text{--}8 \text{ \AA}$ distance between the PS1 catalytic aspartates, although its probability was not high enough to appear as a low-energy state. The discrepancy here could result from potential inaccuracy of the force field parameters and/or still insufficient sampling of the large enzyme-substrate complex. Moreover, as the pKa value of D257 was reasonably close to that of D385 (7.95 vs. 8.80) (**Table S2**), there could be possible proton exchange between the two catalytic aspartates that could not yet be simulated. Furthermore, we could not determine the A β 49/A β 48 ratio quantitatively from the GaMD simulations in this study. While the ratio of AICD50-99 of AICD49-99 was measured at $\sim 1.1 \pm 0.1$ from MS experiments

of the WT APP-bound γ -secretase (**Figure 1**), the ratio between A β 49 and A β 48 produced from WT APP-bound γ -secretase in natural cell lines is $\sim 7:3^{55}$. Nevertheless, the experiments were still proceeded as our focus was to determine the relative differences in the quantities of AICD produced between WT and PS1 FAD mutants. We also mainly examined GaMD free energy profiles between the WT and FAD mutants of PS1....”

Importantly, we did not dismiss anyone’s works here, but we rather tried to be concise as they were described in the Introduction and Discussion sections.

5) What is the evidence that one aspartate needs to be protonated? This is not explained / cited. The use of one protonated aspartate reduces Asp-Asp repulsion and is probably partly responsible for the very short distances seen in this paper, where the two aspartates almost come in contact this is unprecedented to my knowledge, and no experimental or simulated data suggest so – it could easily be an artifact of having protonation on one aspartate. Remember that aspartate needs to be deprotonated to take a proton from the nucleophilic water to enhance its nucleophilicity. This whole topic needs more discussion. A suspicion is that the states here are too compact because the experimental cryo-EM structures are more like 11 Å Asp-Asp distance (should be compared in earnest).

Response: Besides the evidence provided in the answer to the above comment #2, we have included additional references to previous works on why one of the two catalytic aspartates must be protonated in the 2nd paragraph on page 4 as “...The coarse-grained simulations of PS1 as part of the γ -secretase complex illustrated that PS1 was much more likely to be activated when either of the catalytic aspartates was protonated²³. This finding was in good agreement with the proposed mechanism of aspartic proteases, which requires one of the catalytic aspartates to act as an acid²⁴. Hitzengerger and Zacharias observed that the active state of PS1 remained stable even in the absence of a substrate as the direct hydrogen bond between protonated D257, D385, and a water bridge was sufficient to stabilize the active form^{21,25}....”

6) **Table S2.** The authors need to explain why the simulations vary in length, with some mutants simulated for longer and some for less – if they are not treated equally how can you be sure your comparison is not confounded by simulation extent?

Response: We have extended all simulations to the same lengths of 1.5 μ s and updated all the time courses and free energy profiles accordingly. Regardless, the main results and conclusions of this work did not change with the extended simulations.

7) On a similar note (and related to point 2), a direct comparison of WT vs. mutants in context of the intra-system variation (error bars/Precision) seen from the three triplicates is expected – this is normal standard to identify if your conformation tendencies are significantly different in the mutants and WT. For example, you could use bar plots with SD to quantify the distances between Aspartates or to the substrate in WT vs. mutants. Right now, comparisons appear sporadic, which is always prone to selection bias. See Figure S2 and S3: The noise in the graphs make any difference between the systems less significant, and multiple simulations would help to determine if this is the case.

Response: We have made **Figure S1** and described it in the 2nd paragraph on page 8 as “...In particular, the average D257-D385 distances obtained from GaMD simulations were 7.3 ± 1.9 Å for WT, 7.6 ± 1.1 Å for P117L, 8.2 ± 1.6 Å for I143T, 8.7 ± 1.0 Å for L166P, 8.1 ± 1.2 Å for G384A, 9.1 ± 1.2 Å for L435F, and 7.4 ± 1.0 Å for L286V PS1 FAD mutant γ -secretase (**Figure S1**)....”

8) The initial structure and the final representative cluster structures (from equilibrated parts of the trajectories) should be uploaded as PDB files, otherwise it is impossible to understand and reproduce this work and determine the quality and realism of the models applied.

Response: PDB files of the structures were uploaded as part of SI, along with the review response and revised manuscript.

9) Many of the central aspects used in the analysis are drawn from already published work, which has evidently been read but not cited. Such as the monitoring of Asp-Asp and Asp-substrate distances, without reference and comparison to earlier work, the use of names such as “semi closed” conformation is confusing since other groups used the same name or “semi-open” for perhaps other conformation states. All these similar names are confusing to the outsider if not clearly defined by some metric, like e.g., the Asp-Asp-distance or the size of the catalytic pocket, as attempted by the Kepp group.

Response: First, we have explained the reason why we monitored the D257-D385 and D385-L49 distances and cited proper references in the 1st paragraph of page 9 as “...The ϵ -cleavage of APP by γ -secretase can only be carried out when the two PS1 catalytic aspartates are at a suitable distance so that a nucleophilic water molecule can be recruited for the proteolytic reaction through water-bridged hydrogen bonding with the two aspartates^{5,26,30}. Furthermore, the carbonyl group at the cleavage site on APP (residue L49) would form another hydrogen bond between the carbonyl oxygen and protonated carboxylic side chain of catalytic residue D385 in PS1 for proteolysis^{5,26,30}....”

Secondly, to avoid confusion, we replaced the “semi-open” terms as follows:

- 2nd paragraph on page 14: “...In fact, with ~ 7 Å distance between D257–D385, the active site in the “Active” WT conformational state was properly poised for the two catalytic aspartates to recruit a water molecule...”

Meanwhile, the term “semi-closed” was already defined and used as follows:

- 2nd paragraph on page 23: “...In addition, the P117L and L286V PS1 mutants sampled the “I2” state, in which the active site appeared “semi-closed”, with the two catalytic aspartates moving close to each other (**Figures 1E, 1J, and 6C**). Here, a “semi-closed” active site is defined as having a \sim D257–D385 distance between ~ 6 – 6.5 Å²⁶....”

- 2nd paragraph on page 24: “...The presence of these two conformations in its free energy profile indicated that I143T had the ability to “semi-close” the PS1 active site, preventing the APP substrate from being properly located for its ϵ cleavage....”

10) A lot of the results section is way too long and insignificant given the uncertainties in finer details of MD structures. The authors speculate on energies and structures with two decimal precisions, when the real uncertainty is probably in the full kcal/mol unit scale. The authors really

need to get a sense of proportion not to tire the reader with details that are not important. The central part is general agreement with experiment in relevant limits / conditions, show effect of mutations, compare to previous simulations, define clearly and quantitatively the few much chemically important states, forget about small Å displacements and two-decimal energy effects, and then explain better the novelty and significance.

Response: We agree with the reviewer that it is important to balance the descriptions of various details and major findings in the Results section. We have kept only 1 decimal point for energy and distance values in the revised manuscript and remove redundant descriptions about turns and β -bridge secondary structures for APP in the N-terminus.

We have also made further clarifications as in the 1st paragraph on page 7, "...Quantification of the total AICD produced by FAD-mutant γ -secretase revealed significantly decreased ϵ cleavage compared with WT γ -secretase (**Figure 1B**)...", and in the 1st paragraph on page 8, "...Both species of AICD were significantly decreased for all the tested FAD mutants when compared to the WT γ -secretase (**Figure 1B**)...". Our main findings have been summarized starting from the 4th paragraph on page 18 as "...In this work, we have presented dynamic models for cleavage of amyloid precursor protein (APP) by PS1 FAD mutants of γ -secretase, which were consistent with mass spectrometry (MS) and western blotting biochemical experiments. Through the quantifications of the total AICD species produced by WT and PS1 FAD mutant γ -secretase, our biochemical experiments revealed significantly decreased ϵ -cleavages of APP by the PS1 FAD mutants compared to WT γ -secretase³⁹⁻⁴¹. Since the PS1 FAD mutants mostly reduced ϵ -cleavage efficiency, the catalytic efficiency should be reduced, which means lower values of k_{cat}/K_M . The reason the experimental results specifically show reduction in k_{cat} is that they are performed under conditions of substrate saturation. Under these conditions the rate is only determined by the k_{cat} and the concentration of enzyme, the latter which is kept constant. Therefore, a reduced rate of AICD product formation is due to a corresponding decrease in the k_{cat} . GaMD simulations were carried out in parallel to explain the biochemical results in atomistic details. From the 2D free profiles calculated from GaMD simulations, important low-energy conformational states were identified for each simulation system of γ -secretase. The free energy landscapes and low-energy conformational states were explored in detail, which allowed us to deduce the effects of PS1 FAD mutants on the proteolytic activity of γ -secretase. Here, our main conclusion was that the PS1 FAD mutant γ -secretase stabilized the active sites of the enzyme-substrate complexes, which was distinctly different from previous studies, which suggested that PS1 FAD mutants destabilized the enzyme-substrate complexes, causing the earlier releases of longer A β peptides^{10,22,42-45}....". We also compared our findings with previous works as follows:

- 1st paragraph on page 19 as "...Here, our main conclusion was that the PS1 FAD mutant γ -secretase stabilized the active sites of the enzyme-substrate complexes, which was distinctly different from previous studies, which suggested that PS1 FAD mutants destabilized the enzyme-substrate complexes, causing the earlier releases of longer A β peptides^{10,22,42-45}...."

- 2nd paragraph on page 19 as "...Our experimental method has already been validated in another recent study¹⁵. In that study, we quantified all proteolytic events by γ -secretase on C100-Flag substrate with WT and 14 FAD-mutant substrates. For these 15 variants of C100-Flag, the quantification of AICD-Flag using the western blotting method (with C100-Flag itself used as the standard) gave results that were highly consistent with those from LC-MS/MS quantification of small peptide carboxypeptidase coproducts¹⁵. In deducing the production of all A β variants from these data, we found that total AICD equaled total A β in all cases. Moreover, the sums of A β

peptides produced along the A β 40-producing pathway from A β 49 and along the A β 42-producing pathway from A β 48 were equivalent to their corresponding AICD products (AICD50-99 and AICD49-99, respectively)¹⁵. If the quantification of AICD-Flag using C100-Flag as the standard were inaccurate, such close agreement between AICD and A β products would not have been observed. Moreover, while the AICD bands produced from I143T, L166P and L435F were extremely faint, they were visible and within range of the standard curve (stronger than the band of the lowest concentration standard) (**Figure 1B**)....”

- 2nd paragraph on page 20 as “...The experimental effects seen on AICD production with the specific PS1 mutations under study here have also been reported by other groups^{13,17,46-48}. According to Chávez-Gutiérrez et al., AICD production was reduced by the G384A, L166P, and I143T PS1 FAD mutants⁴⁹. Severely compromised γ -secretase activity with the L435F PS1 FAD mutant has been previously reported by several groups^{17,46-48}. For the L286V PS1 FAD mutant, we are only aware of our own previous report on its effect on ϵ cleavage to AICD¹³. In that report, we did not see decreased AICD production vis-à-vis WT enzyme; however, γ -secretase components were overexpressed in Chinese hamster ovary (CHO) cells with endogenous enzyme present, and assays were conducted using isolated membranes, not purified enzyme complexes. Therefore, we favor the results from our current study, which were obtained with purified enzyme and more rigorous quantification of AICD using a standard curve. For the P117L PS1 FAD mutant, we are unaware of any reports on the overall proteolytic activity, only A β 42/40 ratios....”

- 2nd paragraph on page 21 as “...This finding was highly consistent with previous simulation studies, in which the repeated formations of β -strands in several solvent-exposed regions of presenilin were observed^{22,43,51,52}....”

- 1st paragraph on page 23 as “...Nevertheless, it was worth noting that the β 3 strand of APP was formed in all three “Active” low-energy conformations (WT, P117L, and L286V), being consistent with previous studies^{22,43,51,52}....”

- 1st paragraph on page 25 as “...This finding was consistent with that by Chen and Zacharias²⁶, even though their simulations were performed on apo γ -secretase. Chen and Zacharias found that mutation of L435, which was located in close proximity to the active site, to phenylalanine shifted the D257-D385 C γ -distance to larger distances and increased the equilibrium C γ -C γ distance by 0.3 Å²⁶. While our conclusions were identical, the effect could be observed much more clearly with GaMD: the L435F mutation increased the average C γ -C γ distance from 7.3 ± 1.9 Å in WT γ -secretase to 9.1 ± 1.2 Å in the L435F PS1 FAD mutant (**Figure S1**). Furthermore, notable changes in the conformational spaces of PS1-APP interactions were found in all six PS1 FAD mutants, which were consistent with previous experimental and computational results^{10,13,14,16-18,26}....”

- 2nd paragraph on page 26 as “...Our findings were also in good agreement with Chen et al. and others^{17,18,22,26,43,51,52}, even though the effects were clearer due to the enhanced sampling power of GaMD....”

11) Gamma secretase is a very slow enzyme. (k_{cat} for APP ϵ proteolysis ~ 2-6 per hour). Do the effects here translate into a K_M or k_{cat} effect in the enzyme? This really needs to be discussed in contrast to many of the unimportant details that take space now.

Response: We have added the correlation starting from the 2nd paragraph on page 18 as “...Since the PS1 FAD mutants mostly reduced ϵ -cleavage efficiency, the catalytic efficiency should be reduced, which means lower values of k_{cat}/K_M. The reason the experimental results specifically show reduction in k_{cat} is that they are performed under conditions of substrate saturation. Under

these conditions the rate is only determined by the k_{cat} and the concentration of enzyme, the latter which is kept constant. Therefore, a reduced rate of AICD product formation is due to a corresponding decrease in the k_{cat}”

Regarding methodology:

1) The authors should show for some mutations studied before by other labs that their cleavage processing effects are consistent with those of other groups and not lab/protocol dependent, as one can imagine lab specific variations. Always essential to establish agreement and consensus, both to previous simulations and previous experiments.

Response: We have addressed this comment in the response to comment #10 from above.

2) The rationale for using a protonated Asp needs to be explained as it could explain why their ensembles produce much shorter Asp-Asp distances (down to few Å, suggesting almost direct contact between a protonated and non-protonated Asp) than seen experimentally (10-12 Å). In that light, dismissing previous work due to the use of two deprotonated aspartates without explanation of references to the need for one to be protonated is also not enough. The claim that the enzyme cannot be active with two deprotonated Asp seems incorrect (page 4).

Response: We have addressed this comment in the responses to comments #2 and #5 from above.

3) Also previous work studying the gamma secretase cryo-EM structures under various conditions also studied PH effects with PropKa and found results distinct from those here –“only at pH 5 did a protonated state appear for the enzyme complex.”

<https://pubs.rsc.org/en/content/articlelanding/2020/cp/c9cp06723j/unauth>

This needs to be discussed and explained, as the deprotonation may explain a lot of the unusual compact structures in the present work.

Response: Besides the evidence provided in the answer for comment #2, we also included some references to previous works on why one of the two catalytic aspartates must be protonated in the 2nd paragraph on page 4 as “...The coarse-grained simulations of PS1 as part of the γ -secretase complex illustrated that PS1 was much more likely to be activated when either of the catalytic aspartates was protonated²³. This finding was in good agreement with the proposed mechanism of aspartic proteases, which requires one of the catalytic aspartates to act as an acid²⁴. Hitzengerger and Zacharias observed that the active state of PS1 remained stable even in the absence of a substrate as the direct hydrogen bond between protonated D257, D385, and a water bridge was sufficient to stabilize the active form^{21,25}...”

4) The authors forget to tell what they did with the large hydrophilic loop, which is not in the cryo EM structure they start out from, but which is present in the real system and thus experimental / biological relevant data. So, are you comparing apples and berries? The large loop combines TM6 and 7 and could have major influence on modulating the active site, as previous work suggests. The absence of the loop could also reduce hydrophobic repulsion between the NTF and CTF loop parts after maturation which then could also, like the deprotonation, contribute to the very short Asp-Asp distances observed.

Response: Again, this comment was based on the reviewer's inaccurate assumption about the experimental D257-D385 distance. We elaborated on our system setup on pages 29 and 30 as "...The simulation system setup followed a similar protocol as our previous work³⁰.... The large missing hydrophilic loop that connected TM6a and TM7 was not modelled as in our previous studies^{30,31}, which had no significant effects on our final results. In fact, the large missing hydrophilic portion that connects TM6a and TM7 is missing in the cryo-EM structure³⁶, but this region is not conserved and does not contain sites of PS1 FAD mutations. Moreover, Gopal Thinakaran's lab demonstrated years ago that this region is unnecessary for presenilin proteolytic function⁵⁹. This is in contrast to the hydrophobic region of loop 6, which is conserved, critical for function and a domain with many PS1 FAD mutations⁵⁹. The autoproteolytic cleavage of this loop upon assembly of presenilin with the other components of the γ -secretase complex results in the functional protease. The hydrophobic portion of the cleaved loop 6 becomes the TM6a region that is folded into the structure of γ -secretase⁵⁹. The hydrophilic region, now the N-terminus of the presenilin CTF subunit generated by autoproteolysis, is not visible by cryo-EM, even with bound substrate, presumably because it is unstructured and not folded into the active protease complex⁵⁹....."

5) The authors use an MD method that introduces approximations to the sampling by a Gaussian function, but the impact of this approximation is not shown by comparison to a full conventional simulation for this protein. It is relevant to show for gamma secretase specifically because of its very loose conformational states and many (potentially non-Gaussian) modes. This again illustrates that the present simulations are different from, but not necessarily better than previous simulations, yet this comparison and discussion is missing in the paper.

Response: We have performed additional conventional MD (cMD) simulations (1500 ns x 4) on each of four representative APP-bound γ -secretase systems, including the WT, P117L, I143T, and L166P PS1 FAD mutants. The new cMD simulation results compared with GaMD simulations have been described starting from the 3rd paragraph on page 20 of the Discussion section in the revised manuscript as "...We performed four additional 1.5 μ s cMD simulations on each of four representative APP-bound γ -secretase systems, including the WT and the P117L, I143T, and L166P PS1 FAD mutants. The time courses of the D257-D385, D385-V50, D385-L49, and D385-T48 distances calculated from the cMD simulations were plotted in **Figure S15**. 2D free energy profiles of the (D257-D385, D385-L49), (D257-D385, D385-V50), or (D257-D385, D385-T48) distances (**Figure S16**) were calculated and compared with those from GaMD simulations (**Figure 1** and **S17**). For both the cMD and GaMD simulations, the low-energy conformational states calculated from both the D385-V50 and D385-T48 distances matched those calculated from the D385-L49 distances. Moreover, GaMD sampled larger conformational space than the cMD simulations and uncovered additional low-energy conformational states in the WT, I143T, and L166P FAD mutant γ -secretase systems (**Figures 1** and **S17**). In particular, the WT, I143T, and L166P simulation systems visited two ("Inhibited" and "I1"), one ("I4"), and one ("I3") additional low-energy conformational states in the GaMD simulations than in the cMD simulations, respectively (**Figures 1D,F,G** and **S16A,C,D**). In the P117L simulation system, both GaMD and cMD uncovered two low-energy conformational states, i.e., the "Active" and "I2" (**Figures 1E** and **S16B**). These findings demonstrated the enhanced sampling power of GaMD in simulations of large biomolecules such as γ -secretase...."

6) Previous work found that the substrate helix partly unzips when in the pocket – any evidence for this would be relevant to discuss in relation to how the nucleophilic water accesses the peptide bonds consecutively.

Response: In this work, our focus was solely on the effects of PS1 FAD mutants on the ϵ -cleavage of APP by γ -secretase. The effects of the partial substrate helix unzipping could be found in our recent publications: [10.1021/acscentsci.0c00296](https://doi.org/10.1021/acscentsci.0c00296), [10.1021/jacs.1c10533](https://doi.org/10.1021/jacs.1c10533)

7) The seven lines of conclusion on page 22-23 are very vague and it is not clear what is old and what is new, compared to previous insights. After going through all the above I believe it will be easier for the authors to write a conclusion that provides strong information on what actually is gained at the molecular level from this paper.

Response: We have strengthened our conclusion in the 2nd paragraph on page 26 “...In conclusion, we have presented the dynamic models for cleavage of amyloid precursor protein (APP) by PS1 FAD mutants of γ -secretase, which were consistent with mass spectrometry (MS) and western blotting biochemical experiments. Our findings were also in good agreement with Chen et al. and others^{17,18,22,26,43,51,52}, even though the effects were clearer due to the enhanced sampling power of GaMD. First, we found that the PS1 FAD mutants confined the active site in PS1 and APP substrate. Second, the PS1 FAD mutants were found to reduce γ -secretase proteolytic activity by hindering APP residue L49 from proper orientation in the active site and/or disrupting the distance between the catalytic aspartates. Our findings here provided mechanistic insights into how PS1 FAD mutants affect structural dynamics and enzyme-substrate interactions of γ -secretase and APP....”

In summary, the paper’s work is substantial but also not very novel and the presentation is very incomplete, and several points of the methodology need substantial further attention. I recommend major revision and probably submission to a more technical journal although Commun. Biol is also possible if the advice above is followed and a lot of less relevant discussion on pages 8-22 is replaced with more direct assertive comparison to experimental cryo-EM structures and previous simulation work, showing the key molecular insights in better focus.

Response: We hope that our responses above clearly demonstrate that this study is indeed novel, providing new insight into how FAD PS1 mutations affect endoproteolysis of APP substrate, and that the study is complete, including sufficient description of the methodology to allow replication by other investigators.

Minor points:

1) it is very hard to see the differences claimed in **Figure S5**

Response: The old **Figure S5**, now **Figure S6**, showed the overall pictures of the PS1 domains in the intermediate low-energy conformational states compared to “Active” WT conformation and only served as an overview for what would be described in specific details in **Figures S7-S9** (detailed differences of low-energy conformational states of the simulated systems). We thought it would be the best to give the reader an overview first before going into the specific details.

2) the use of names for conformations is complicated and a lot of this discussion could be simplified.

Response: To avoid confusion, we replaced the “semi-open” terms as follows:

- 2nd paragraph on page 14: “...In fact, with $\sim 7 \text{ \AA}$ distance between D257–D385, the active site in the “Active” WT conformational state was properly poised for the two catalytic aspartates to recruit a water molecule...”

Meanwhile, the term “semi-closed” was already defined and used as follows:

- 2nd paragraph on page 23: “...In addition, the P117L and L286V PS1 mutants sampled the “I2” state, in which the active site appeared “semi-closed”, with the two catalytic aspartates moving close to each other (**Figures 1E, 1J, and 6C**). Here, a “semi-closed” active site is defined as having a \sim D257–D385 distance between $\sim 6\text{--}6.5 \text{ \AA}$...”

- 2nd paragraph on page 24: “...The presence of these two conformations in its free energy profile indicated that I143T had the ability to “semi-close” the PS1 active site, preventing the APP substrate from being properly located for its ϵ cleavage....”

Reviewer #3 (Remarks to the Author):

It is very important to understand how FAD mutants of PS1 affect structural dynamics of gamma-secretase complex and substrate, in terms of Abeta42 production. In this study, authors performed structural simulation study and biochemical analyses on several FAD mutants of PS1 and APP derived substrate. Although one may consider that simulation study is just a prediction, it helps us to gain insights into how this enzyme produces Abeta42. This reviewer has several comments on this manuscript.

1. Please include the following references in the introduction part of tricarboxypeptidase trimming, because this is the 1st paper to demonstrate it: gamma-Secretase: successive tripeptide and tetrapeptide release from the transmembrane domain of beta-carboxyl terminal fragment. Takami M, Nagashima Y, Sano Y, Ishihara S, Morishima-Kawashima M, Funamoto S, Ihara Y. *J Neurosci*. 2009 Oct 14;29(41):13042-52. doi: [10.1523/JNEUROSCI.2362-09.2009](https://doi.org/10.1523/JNEUROSCI.2362-09.2009).

Response: We have cited the reference in the last sentence of 1st paragraph on page 6 of the Introduction as “...The simulation findings were also highly consistent with biochemical experimental data^{31,32}...”

2. Quantification of AICD No.1: It seems that AICD bands on I143T, L166P, and L435F are not detected at all (Fig. 1B). If so, these are out of range of the standard curve? This reviewer wonders how the authors quantified AICD levels.

Response: We have clarified this concern in the 1st paragraph on page 20 as “...Moreover, while the AICD bands produced from I143T, L166P and L435F were extremely faint, they were visible and within range of the standard curve (stronger than the band of the lowest concentration standard) (**Figure 1B**)....”

3. Quantification of AICD No.2: The authors used C100, as an authentic standard to quantify AICD concentration. However, this reviewer wonders whether retention level of AICD on PVDF

membrane is same as that of C100, or not. Because AICD is much smaller than C100, so retention level of AICD on the membrane is less. The best way to quantify AICD on the membrane is to use AICD as authentic standard, not C100.

Response: Although this is a reasonable concern, we have already validated this method in Devkota et al. JBC 2021. We have added the clarification starting from the 2nd paragraph on page 19 as “...Our experimental method has already been validated in one other recent study¹⁵. In that study, we quantified all proteolytic events by γ -secretase on C100-Flag substrate with WT and 14 FAD-mutant substrates. For these 15 variants of C100-Flag, the quantification of AICD-Flag using the western blotting method (with C100-Flag itself used as the standard) gave results that were highly consistent with those from LC-MS/MS quantification of small peptide carboxypeptidase coproducts¹⁵. In deducing the production of all A β variants from these data, we found that total AICD equalled total A β in all cases. Moreover, the sums of A β peptides produced along the A β 40-producing pathway from A β 49 and along the A β 42-producing pathway from A β 48 were equivalent to their corresponding AICD products (AICD50-99 and AICD49-99, respectively)¹⁵. If the quantification of AICD-Flag using C100-Flag as the standard were inaccurate, such close agreement between AICD and A β products would not have been observed....”

4. This reviewer wonders whether 2D free energy profiles with distance between PS1 Asp385 and C100 Val50 are similar to Fig. 1E-J. Any reason why no simulation with Asp385-Val50?

Response: We calculated the time courses for the D385-V50 distances for both the GaMD and new cMD simulations in **Figures S2-S5** and **S15**. The 2D free energy profiles of the D257-D385 and D385-V50 distances were calculated in **Figures S16** and **S17** for the cMD and GaMD simulations, respectively. We described our new results in the 2nd paragraph on page 20 of the Discussion section as “...We performed four additional 1.5 μ s cMD simulations on each of four representative APP-bound γ -secretase systems, including the WT and the P117L, I143T, and L166P PS1 FAD mutants. The time courses of the D257-D385, D385-V50, D385-L49, and D385-T48 distances calculated from the cMD simulations were plotted in **Figure S15**. 2D free energy profiles of the (D257-D385, D385-L49), (D257-D385, D385-V50), or (D257-D385, D385-T48) distances (**Figure S16**) were calculated and compared with those from GaMD simulations (**Figure 1** and **S17**). For both the cMD and GaMD simulations, the low-energy conformational states calculated from both the D385-V50 and D385-T48 distances matched those calculated from the D385-L49 distances. Moreover, GaMD sampled larger conformational space than the cMD simulations and uncovered additional low-energy conformational states in the WT, I143T, and L166P FAD mutant γ -secretase systems (**Figures 1** and **S17**). In particular, the WT, I143T, and L166P simulation systems visited two (“Inhibited” and “I1”), one (“I4”), and one (“I3”) additional low-energy conformational states in the GaMD simulations than in the cMD simulations, respectively (**Figures 1D,F,G** and **S16A,C,D**). In the P117L simulation system, both GaMD and cMD uncovered two low-energy conformational states, i.e., the “Active” and “I2” (**Figures 1E** and **S16B**). These findings demonstrated the enhanced sampling power of GaMD in simulations of large biomolecules such as γ -secretase....”

5. This reviewer is curious to know when C100 gets cleaved in time course studies in Fig. 5.

Response: The chemical reaction of C100 cleavage by γ -secretase could not be modeled unfortunately by Molecular Dynamics simulations alone. It would require Quantum Mechanics calculations, which is beyond the scope of this study. Instead, we rather captured all the necessary conformational details of the PS1 active site, APP, and nucleophilic water poised for the substrate cleavage reaction.

REVIEWERS' COMMENTS:

Reviewer #1 (Remarks to the Author):

The authors properly addressed my comments and my general feeling is that the manuscript is now ready for publication.

Reviewer #2 (Remarks to the Author):

The authors have done an excellent job revising the paper according to my suggestions and responded to my concerns. I appreciate the authors work on this and their extensive revision based on my critical first report. I can accept the revised paper for publication.

A few minor points:

While I accept the kinetic interpretation of the present MD data this is in disagreement with some previous interpretations – this is interesting in science, as it gives us more opportunity to determine which models are right (stabilization mechanism as here or destabilization of the enzyme substrate complex as in the FIST model), so it opens up a door to more studies and discussions.

It could perhaps reduce confusion if it was stated more clearly early in the text that the Asp-Asp distance is not alpha carbon-based as in previous work but based on C-gamma atoms – at least this explains a lot of the confusion I had in my original report, as I wrongly simply assumed it was Calpha.

A lot of new technical detail and text was added to the discussion as a result of revision in response to the reviewers. Given the very long text and some points less central to the key analysis (the confinement mechanism and the action of the mutants via L49) I still recommend shortening the discussion to focus on the most important points to make the paper more suitable for the journal's audience.

Except from this, I recommend publication and do not suggest a need to see a 2nd revised version.

Reviewer #3 (Remarks to the Author):

The authors' responses are quite adequate. This reviewer believes this is an important study on understanding how FAD mutations influence APP processing.

We thank the editor and reviewers for the praises and acceptances of the revised manuscript. The remaining minor comments are addressed pointwise below.

REVIEWERS' COMMENTS:

Reviewer #1 (Remarks to the Author):

The authors properly addressed my comments, and my general feeling is that the manuscript is now ready for publication.

Reviewer #2 (Remarks to the Author):

The authors have done an excellent job revising the paper according to my suggestions and responded to my concerns. I appreciate the authors work on this and their extensive revision based on my critical first report. I can accept the revised paper for publication.

A few minor points:

While I accept the kinetic interpretation of the present MD data this is in disagreement with some previous interpretations – this is interesting in science, as it gives us more opportunity to determine which models are right (stabilization mechanism as here or destabilization of the enzyme substrate complex as in the FIST model), so it opens up a door to more studies and discussions.

It could perhaps reduce confusion if it was stated more clearly early in the text that the Asp-Asp distance is not alpha carbon-based as in previous work but based on C-gamma atoms – at least this explains a lot of the confusion I had in my original report, as I wrongly simply assumed it was Calpha.

Response: We have explicitly stated that we used the C_γ-atom distance between PS1 residues D257 and D385 as a reaction coordinate in the 1st paragraph on page 9 as “...Therefore, the distance between the C_γ atoms of catalytic aspartates D257 and D385 in PS1 and the distance between PS1 residue D385 (protonated oxygen) and APP residue L49 (carbonyl oxygen) were calculated from the GaMD simulations and plotted in **Figures S2–S5**. They were used as reaction coordinates to calculate two-dimensional (2D) potential mean force (PMF) free energy profiles to characterize the effects of PS1 FAD mutations on γ -secretase activation for ϵ cleavage of APP (**Figure 1**)....”

A lot of new technical detail and text was added to the discussion as a result of revision in response to the reviewers. Given the very long text and some points less central to the key analysis (the confinement mechanism and the action of the mutants via L49) I still recommend shortening the

discussion to focus on the most important points to make the paper more suitable for the journal's audience.

Response: We have gone through the Discussion section again and concluded that many readers might have similar questions as the reviewers about the study. Therefore, we decided to keep the Discussion section as it is right now.

Except from this, I recommend publication and do not suggest a need to see a 2nd revised version.

Reviewer #3 (Remarks to the Author):

The authors' responses are quite adequate. This reviewer believes this is an important study on understanding how FAD mutations influence APP processing.